# The regulatory landscape of *Arabidopsis thaliana* roots at single-cell resolution

Michael W. Dorrity [1], Cristina M. Alexandre[1], Morgan O. Hamm[1], Anna-Lena Vigil[2], Stanley Fields [1,3], Christine Queitsch [1✉] & Josh T. Cuperus [1✉]

The scarcity of accessible sites that are dynamic or cell type-specific in plants may be due in part to tissue heterogeneity in bulk studies. To assess the effects of tissue heterogeneity, we apply single-cell ATAC-seq to *Arabidopsis thaliana* roots and identify thousands of differentially accessible sites, sufficient to resolve all major cell types of the root. We find that the entirety of a cell's regulatory landscape and its transcriptome independently capture cell type identity. We leverage this shared information on cell identity to integrate accessibility and transcriptome data to characterize developmental progression, endoreduplication and cell division. We further use the combined data to characterize cell type-specific motif enrichments of transcription factor families and link the expression of family members to changing accessibility at specific loci, resolving direct and indirect effects that shape expression. Our approach provides an analytical framework to infer the gene regulatory networks that execute plant development.

[1] Department of Genome Sciences, University of Washington, Seattle, WA, USA. [2] School of Life Sciences, University of Nevada, Las Vegas, NV, USA. [3] Department of Medicine, University of Washington, Seattle, WA, USA. ✉email: queitsch@uw.edu; cuperusj@uw.edu

Single-cell genomics allows an unbiased sampling of cells during development with the potential to reveal the order and timing of gene regulatory and gene expression events that specify cell identity and lineage. An ideal system to test the ability of single-cell genomics to provide insights into development is the *Arabidopsis thaliana* root: along its longitudinal axis, a single, radially-symmetric root captures developmental trajectories for several cell types. Approaches in this organism have included single-cell RNA-seq to transcriptionally profile individual root cell types along this developmental axis[1–6], and with respect to their ploidy.

Studies of chromatin accessibility in samples enriched for specific plant cell types have revealed: (i) the existence of cell type-specific regulatory elements; (ii) the relative scarcity of such elements compared to their prevalence in animals or humans; (iii) the expected enrichment of transcription factor binding sites within these elements; and (iv) a higher frequency of dynamic regulatory elements upstream of environmentally-responsive genes than constitutively expressed genes[7,8]. Although the correlation between chromatin accessibility and nearby gene expression is generally weak in both plants and animals[9], this correlation improves for regulatory elements that show dynamic changes in chromatin accessibility, for example in response to an environmental stimulus or developmental signal[7,9–11]. In contrast to animals, however, the majority of chromatin-accessible sites in plants show little change across tissues, conditions, or even genetic backgrounds, raising the possibility that cell and tissue identity is less rigidly engrained in the chromatin landscape in plants than in animals[7]. Alternatively, cell type-specific regulatory elements and gene expression in plants may have been obscured by tissue heterogeneity in bulk tissue studies.

Cell type-specific chromatin-accessible landscapes are also of interest for addressing other fundamental biological questions. General transcription decreases along a cell type's developmental trajectory, while expression of cell type-specific genes increases[2,12,13], in agreement with Waddington's predictions on epigenetic landscapes[14]. In the *A. thaliana* root, the increasing maturity of certain cell layers is accompanied by endoreduplication. The presence of additional gene copies may contribute to the observed increase in the expression of cell type-specific genes; alternatively, the initial gene copies may increase their transcription. Although endoreduplication is a common mechanism to regulate cell size and differentiation in plants and some human and animal tissues[15–17], the influence of this phenomenon on gene regulation and expression has been largely overlooked. In plants, endoreduplication generally enhances transcription[17,18], in particular of cell wall-related genes[19] and genes encoding ribosomal RNA[20], hinting at a role for this process in driving increased translation.

Here, we provide single-cell resolution maps of open chromatin in the *A. thaliana* root to address the issue of tissue heterogeneity and to detect likely endoreduplication events. We use a droplet-based approach to profile over 5000 nuclei for chromatin accessibility, and identify 8000 regulatory elements that together define most cell types of the root. We describe an analytical framework that links patterns of open chromatin with transcriptional states to predict the identity, function, and developmental stage of individual cells in the *A. thaliana* root. We integrate the single-cell ATAC-seq (scATAC-seq) data with published single-cell RNA-seq (scRNA-seq) profiles of the same tissue to obtain automated annotations of cells in our scATAC-seq data. Using the integrated dataset, we link individual cells from our scATAC-seq data with their nearest neighbors in scRNA space to define relative developmental progression, level of endoreduplication, and the genes differentially expressed in these nearest neighbors. This approach allows the identification of three distinct developmental states of endodermis cells, which had escaped detection using scRNA-seq alone. Using scATAC-seq data integrated scRNA-seq data, we predict individual members of large transcription factor families that play a role in epidermis development, pinpointing individual regulatory events that link peak accessibility and transcription factor expression in these cells. The combination of binding motifs, transcription factor expression, and chromatin accessibility provides a basis for predicting the gene regulatory events that underlie development.

## Results

**scATAC-seq identifies known root cell types.** We first asked if ATAC-seq profiles at the single-cell level were capable of capturing known root cell types. We profiled 5283 root nuclei, at a median of 7290 unique ATAC inserts per nucleus. A high fraction of these inserts occurred in one of the 22,749 open chromatin peaks (FRIP score = 0.71) based on pseudo-bulk peak calling (Cellranger v3.1, 10× Genomics); this fraction is similar to that seen in high-quality bulk accessibility studies (Supplementary Fig. 1A, B)[9]. Furthermore, the scATAC assay detected 1794 peaks that had not been observed at appreciable levels in bulk ATAC-seq. We used UMAP dimensionality reduction of the peak by cell matrix to build a two-dimensional representation grouping of cells with similar accessibility profiles (Fig. 1a). Subsequent cluster assignment by Louvain community detection identified nine distinct cell clusters[21]. Across all cells, we identified 4389 peaks (ranging from 307 to 1993 per cell type) with significant differential accessibility, suggesting that around 20% of all accessible sites contain some information on cell type (Supplementary Data 1). Though only 16% (707/4389) of cell type-specific peaks were found to be distal, or greater than 400 base pair from the nearest gene, this was greater than the fraction expected by chance. Only 9.4% (2159/22,749) of all peaks were distal, suggesting that these distal peaks are slightly (1.7×) enriched for regulatory sites that define cell identity. To assign cell type annotations to each of these clusters, we generated "gene activity" scores that sum all ATAC inserts within each gene body and 400 bp upstream of its transcription start site. This approach rests on the assumption that a chromatin-accessible site in the compact *A. thaliana* genome tends to be associated with regulation of its most proximal gene[22]. While this assumption may not hold universally, gene activity scores offer the advantage of allowing a direct comparison to bulk ATAC-seq and single-cell RNA-seq datasets through a matched feature set. In this way, we identified genes whose accessibility signal specifically marks each cell cluster. We visualized peaks with cell type-specific accessibility by grouping cells of a similar type and "pseudo-bulking" their insert counts at each position in the genome (Fig. 1b). Bulk and cell type-specific ATAC signals are similar to those obtained in prior whole tissue and cell type enrichment-based ATAC-seq studies for the root (Supplementary Fig. 1B–D)[11].

We used comparisons to tissue-specific genes that were identified from single-cell RNA-seq studies of the *A. thaliana* root to assign a cell type to each cluster defined by ATAC markers from "gene activity" scores[2,5,6]. We identified 210 genes with unique accessibility patterns across all cell types (Supplementary Data 2); FRIP scores, fragment lengths, and total read counts did not vary greatly across cell types (Supplementary Fig. 1E–G). For each cell type, the median number of genes with tissue-specific accessibility was 20 (range 5–53) (Fig. 1c). This small number of genes is consistent with earlier studies that show few open chromatin sites that define cell type identity in *A. thaliana*[7,23]. Although thousands of differentially accessible sites have been found across tissue types[7], accessibility differences between more closely related cell types remain largely unexplored, with the

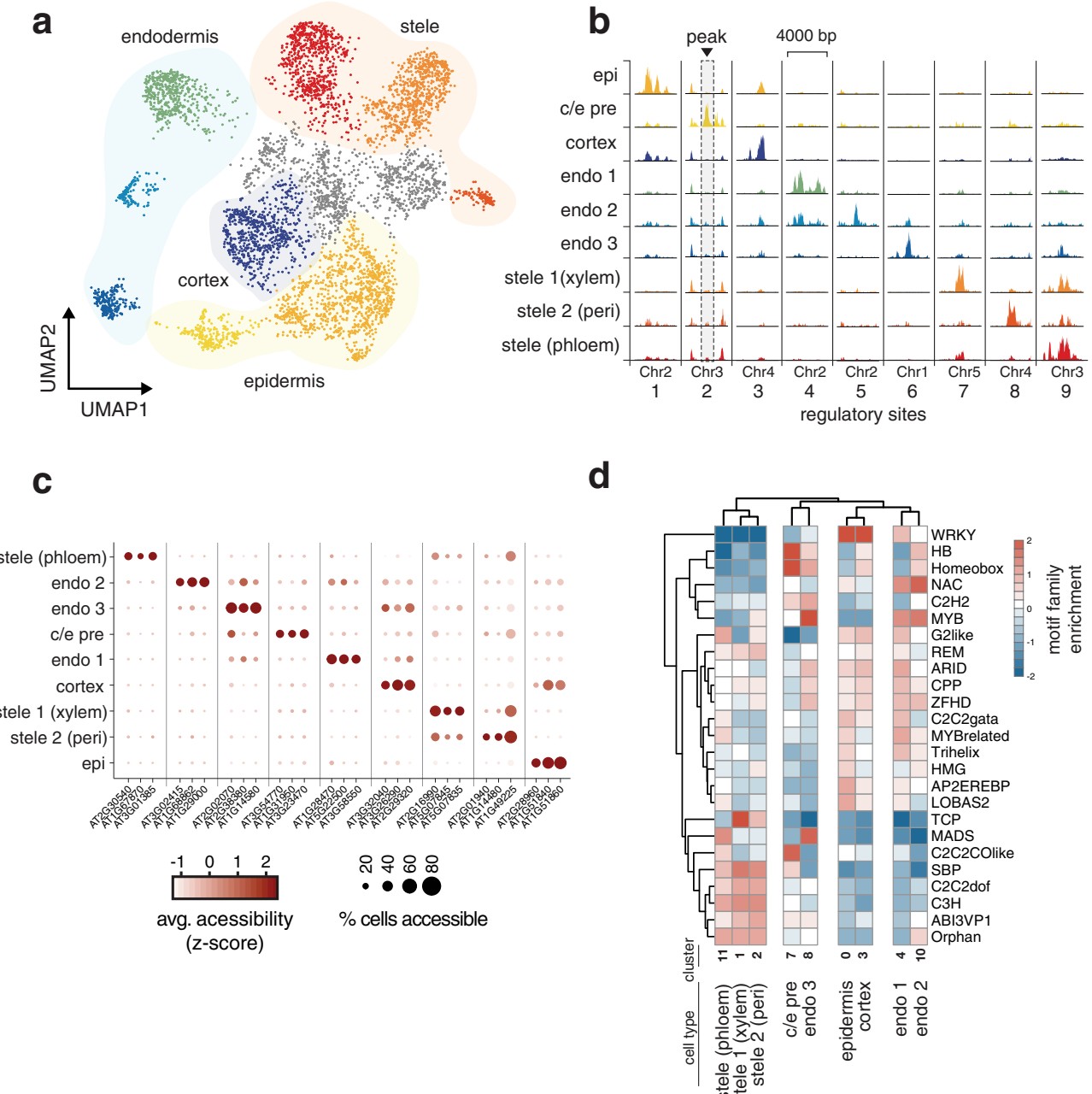

**Fig. 1 scATAC-seq identifies known root cell types. a** UMAP dimensionality reduction plot of root cells using peak-level scATAC data. Cells are colored according to Louvain clusters, and broad tissue types are indicated with transparent shading. **b** Pseudo-bulked peak tracks generated by combining ATAC data from all cells within a cluster. Each column represents a single locus in the genome that shows cell type-specific accessibility; each row represents a cell type, and each column shows an example marker peak for each type. Colors match those in previous panel. A cluster residing between the epidermis and endodermis clusters, with expression of markers from both cell types (Supplementary Fig. 2B, C) was given the label "c/e pre" (precursor of cortex/endodermis, second row), and epidermis was shortened to "epi". **c** Dotplot showing marker genes for each cell type cluster. Each column represents a single gene's activity score, the summed accessibility of its gene body and promoter sequence (−400 bp from transcription start site). The color of each dot indicates the magnitude of accessibility and the size of each dot represents the fraction of cells in each cell type showing accessibility at that gene. **d** Heatmap showing the predicted effect, across all peaks, of motifs from each *Arabidopsis* transcription factor family on cell type-specific accessibility. Darker shades of red indicate that presence of the motif is correlated with increased accessibility in that cell type, whereas shades of blue indicate that the motif is anti-correlated with accessibility. The mean effect all transcription factors within a given family are shown as rows, and each column represents a cell type. Source data for UMAP projection and cell annotations (**a**) are provided as a Source Data file.

exception of root hair vs. non-hair, in which few differences were found[7,11]. These differences, uncovered using a cell-enrichment based technology[11], were replicated in the epidermal cells identified in our scATAC assay (Supplementary Fig. 1C, D). For three cell clusters (959 cells, or 18% of cells), we could not identify a coherent set of a markers and therefore could not

annotate them (gray points, Fig. 1a). However, all other cell clusters were manually annotated and corresponded to the major cell layers of the root (Supplementary Fig. 2A): outer layers including epidermis cortex, and a precursor of endodermis and cortex (ec pre); endodermal layers comprised of three distinct types (endo 1, 2, and 3); and the stele comprised of two main

types along with a phloem type (stele phloem). Several traditional marker genes were used to facilitate annotation of root cell types (Supplementary Fig. 2B–D), as were marker genes identified in previous scRNA-seq studies (Supplementary Data 3). In general, scATAC marker genes did not show a strong overlap with RNA-based marker genes. Endodermis cells were an exception, as several of their scATAC marker genes (AT3G32980, AT1G61590, AT1G14580, AT3G22600, and AT5G66390) were also found to be marker genes in single-cell RNA-seq studies[24]. While this lack of overlap makes annotation more challenging, it is consistent with the reported weak correlation of chromatin accessibility with gene expression[23,25]. Moreover, the finding that expression levels are not precisely predicted by nearby accessible sites suggests that accessibility can add orthogonal information about cell identity to further stratify cell types into distinct sub-types.

**Sequence motifs of transcription factor families associate with cell type-specific sites of open chromatin.** Accessibility at regulatory sites is driven by transcription factor binding and modification of local chromatin[26]. We examined if any of the cell type-specific accessible sites were associated with the presence of transcription factor binding motifs. To do so, we used a set of representative motifs for all *A. thaliana* transcription factor families and nearly every individual transcription factor[27] to tally these motif counts within all 21,889 peaks in the full scATAC-seq dataset to build a peak-by-motif matrix. As each peak can be described in terms of its relative accessibility in each of the identified cell types, we performed a linear regression for each motif to test for significant association of accessibility and motif presence. Relative accessibility values were calculated by first pseudo-bulking all peak counts by cell type, and then normalizing these cell type-specific peak accessibility scores to a background peak accessibility of all cells pooled together. By testing the association of motif counts and cell type-specific accessibility, we identified transcription factor binding motifs whose presence was correlated with higher accessibility in each cell type. However, because motif sequence content for individual transcription factors is redundant, we computed means across each transcription factor family.

We found significant associations with motifs from at least one transcription factor family in all cell types (Fig. 1d). For example, relative chromatin accessibility in epidermal cells was strongly associated ($q$-values ranging from $1e-24$ to $1e-133$) with the presence of motifs from the WRKY transcription factor family; this family includes *TTG2*, which, along with *TTG1* and *GL2*, has an important roles in atrichoblast fate in the epidermis[28]. Furthermore, the effects of each motif family on relative accessibility was sufficient to hierarchically cluster cell types according to broad tissue classes (Fig. 1d). Based on similarities in motif associations, hierarchical clustering grouped all stele clusters (1, 2, and 11), epidermis and cortex (clusters 0 and 3), two endodermis clusters (4 and 10), and another endodermis cluster with epidermal precursor cells (clusters 7 and 8). That motif associations alone can distinguish among clusters and group similar ones together provides an independent verification of the cell type-specific nature of the chromatin-accessible sites detected in the scATAC-seq data.

**Integration of scATAC-seq and scRNA-seq data improves cell type annotation.** Because scATAC-seq data both identified known root cell types and provided cell identity assignments not identifiable through scRNA-seq, we addressed whether combining these two datasets results in additional insights than what could be gained from either alone. We first addressed whether both data types could be embedded in the same low-dimensional space in a manner that maintains the cell identities defined by both scATAC-seq and scRNA-seq. Such embedding assumes that the underlying cell identities represented in each dataset are similar. Although the root tissue sampled for our scATAC-seq experiment was not identical to that used in previous scRNA-seq experiments, we expected that the same major cell types were sampled in both types of experiments. Moreover, the data generated by both methods share "gene" as a feature, i.e., accessibility near or within a given gene; expression of a given gene.

We used the anchor-based multimodal graph alignment tool from the Seurat package to find nearest-neighbor scRNA-seq matches for each cell in the scATAC-seq data[29,30]. In short, the tool identifies representative features (shared "anchor" genes in our case) in each dataset and looks for underlying correlation structure of those features to group similar cells in a co-embedded space. We plotted all cells within the resulting co-embedded space with cell type labels from each dataset separately. Cells derived from scRNA-seq and scATAC-seq experiments were well mixed (Fig. 2a). Moreover, we found that cells of the same type were co-localized independent of the source data (Fig. 2b, c), though some separation by data type was apparent, likely owing to the imputation step of dataset integration[29]. This result suggests that RNA and ATAC signals, which are only poorly correlated in bulk studies, are capable of grouping cell identities when determined in individual cells of a complex tissue. We further used this co-embedded space to refine our earlier manual cell type annotations by transferring labels of neighboring scRNA cells onto the scATAC cells (Supplementary Fig. 3A, B); while most of these labels matched, the greatest number of mismatches was seen in endodermis sub-type 1. The transferred labels matched our manual annotations, and, in the case of epidermal cells, allowed us to separate a single scATAC cluster into hair and non-hair cells (Fig. 2a and Supplementary Fig. 3A, B). Furthermore, this co-embedded space was additionally used to transfer quantitative metrics and gene expression values derived from scRNA-seq data (Supplementary Fig. 3C). The three distinct scATAC clusters that were assigned an "endodermis" label with this approach are a striking example of scATAC data yielding, within a single cell type, greater stratification of "types" than the generally richer scRNA data.

**Epidermal cell layers show increased levels of endoreduplication.** scATAC-seq data can potentially provide insight into DNA copy number and its impact on gene regulation, but this potential has not been thoroughly explored. DNA copy number is of special relevance in the *A. thaliana* root, as each cell layer undergoes different rates of endoreduplication[19]. In a diploid cell, a single accessible locus tends to show 1 or 2 transposition events. In polyploid cells with higher DNA copy number, a single accessible locus could show 4, 8, or even 16 transpositions. Therefore, cells containing a large number of peaks with >1 transposition event are likely to represent endoreduplicated cells. To identify such cells, we classified each cell by the mean number of cuts it contained per peak and examined the distribution of this metric, accounting for differences in total UMI counts (see "Methods" section), to draw a threshold above which cells were classified as likely endoreduplicated (Supplementary Fig. 4A, B). We found the expected trend of higher endoreduplication in the outermost cell files, with reduced prevalence in the stele (Supplementary Fig. 4C).

We then used a second method to identify endoreduplicated cells with a transcriptional signature. Instead of relying on the number of transpositions in the accessibility data directly, we instead leveraged the dataset integration described above (Supplementary Fig. 3C) to transfer scRNA-seq-based annotations to the

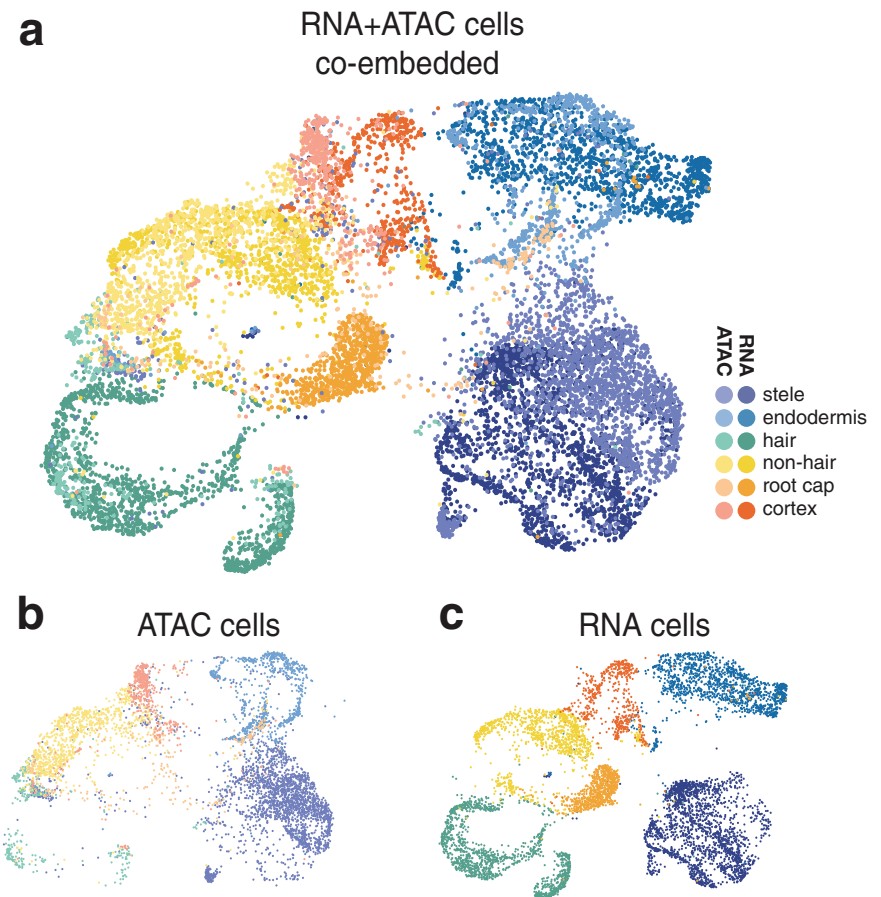

**Fig. 2 scATAC-seq data can be integrated with scRNA-seq data to identify cell types. a** UMAP co-embedding of root scATAC cells alongside root scRNA cells[5]. Cells are colored by broad tissue type, with scATAC cells colored in lighter shades and scRNA cells in darker shades. **b** UMAP from **a**, but showing only cells from the scATAC-seq experiment; **c** shows only cells from the scRNA-seq experiment.

cells in our scATAC experiment. To identify endoreduplicated cells in scRNA-seq data, we used a published set of marker genes, sub-setting the top 250 markers for each ploidy level to generate signature scores for 2n, 4n, 8n and 16n ploidies[19]. With these scores, we predicted endoreduplicated cells by calculating, for each cell, the ratio of the 8n signature relative to the diploid signature. Similar to the accessibility-based metric, this transcription-based approach identified endoreduplicated root cells in the expected pattern, with higher fractions in the epidermis cell layer and diminished levels in the stele (Supplementary Fig. 4D, E). We found these two methods of identifying endoreduplicated cells to be concordant (Supplementary Fig. 4F), but because the accessibility-based classification was less quantitative, we used the transcriptionally-based metric in subsequent analyses. This metric captured an abundance of tetraploid xylem cells in the stele (Supplementary Fig. 4E), consistent with previous findings[19].

**scATAC-seq captures three distinct endodermis types representing different developmental stages.** We dissected the three endodermis clusters in greater detail using three approaches: (i) by identifying differentially accessible sites among sub-types; (ii) by aligning these sub-types to scRNA-seq data that have been annotated for endoreduplication and developmental progression; and (iii) by determining differentially expressed genes in the nearest-neighbors to each of these endodermis sub-types in scRNA-seq space (Fig. 3a).

We identified few differentially accessible genes (adjusted $p$-value < 0.05 and at least 2-fold change in accessibility) in each endodermis sub-type: 25 for the first sub-type, 24 for the second,

and 17 for the third (Fig. 3a). The low number of associated genes precluded gene set enrichment analyses, but genes uniquely accessible in sub-type 1 included transcription factors *MYB85* (AT4G22680) and *NAC010* (AT1G28470) as well as genes involved in suberization (*FAR1*, *FAR4*, and *FAR5*)[31]. Endodermis sub-type 2 showed increased accessibility at *HIPP04* (AT1G2900), encoding a heavy metal-associated protein, *ANAC038* (AT2G24430), and phenylpropanoid metabolism genes[32]. Endodermis sub-type 3 showed strong accessibility at the *BLUEJAY* (AT1G14580) locus encoding a C2H2 transcription factor implicated in endodermis differentiation (Fig. 3b)[33], as well as *MYB122* (AT1G74080) and other genes for phenylpropanoid biosynthesis (*PER22*, *PER32*, *PER72*, and *BGLU32*)[32]. We addressed whether these differentially-accessible genes show different expression patterns in endodermis cells in scRNA-seq space by mapping expression of each gene onto a subclustered set of endodermis cells combined from several scRNA-seq studies of the *A. thaliana* root[2–6]. The small set of marker genes identified for each scATAC sub-type showed no consistent expression pattern in the scRNA-seq data (Supplementary Fig. 5A), suggesting that other features distinguished these three sub-types.

Structure within two-dimensional embeddings of scRNA-seq and scATAC-seq data derived from developing tissues is often associated with differences in developmental progression or other asynchronous processes like the cell cycle. Furthermore, root tissue has the unique feature of being highly endoreduplicated, which could also account for differences among the sub-types. To assess whether the endodermal sub-types were associated with these features, we added annotations for developmental progression,

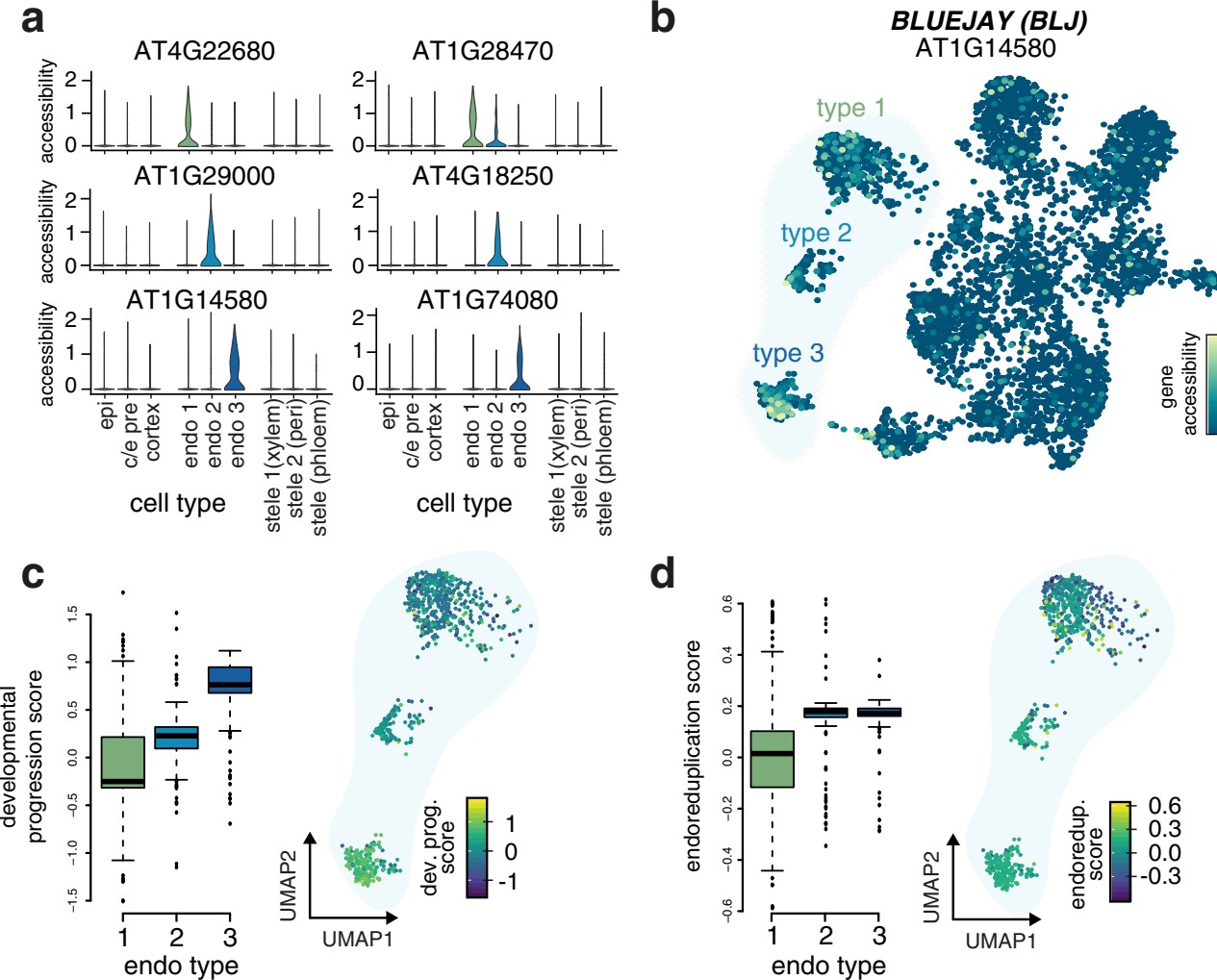

**Fig. 3 scATAC-seq identifies distinct sub-types of endodermal cells. a** Violin plots showing specific patterns of accessible genes that mark each endodermal sub-type. Two examples are given for each endodermal sub-type, with gene-level accessibility scores indicated for all other cell types. **b** UMAP of all cells colored by accessibility of the *BLUEJAY* gene, which marks endodermal type 3; corresponding violin plot for this gene in lower left panel in **a**. **c** Boxplot showing an increase in median developmental progression of each endodermal sub-type, as determined by average transcriptional complexity in the nearest 25 scRNA neighbors of each scATAC cell in the co-embedded representation from Fig. 2a; right inset shows UMAP of endodermal cells with each cell colored by the average developmental progression of its scRNA neighbors, mirroring the gradual increase seen in left panel. Boxplots are generated using values from individual endodermis cells (early $n = 489$ cells, mid $n = 141$ cells, late $n = 225$ cells); whiskers represent 1.5 times the interquartile range of the data, the box represents the interquartile range, and the horizontal line in the box represents the median. **d** Boxplot showing an increase in median levels of endoreduplication across endodermal sub-types, ascertained as in **c**, but instead using a gene expression signature of endoreduplication; right inset shows UMAP of endodermal cells with each cell colored by the average endoreduplication score of its scRNA neighbors, with highest levels seen in endodermal sub-types 2 and 3 (number of cells, $n$, identical to previous panel).

endoreduplication and cell cycle to the combined root scRNA-seq data and used data integration (as in Fig. 2) to test whether cells from the endodermal sub-types were associated with any of these features (Supplementary Fig. 3C).

We assessed developmental progression with two orthogonal methods: (i) correlation with published bulk expression data taken along longitudinal sections of the root[1]; and (ii) a modified measure of loss in transcriptional diversity (see "Methods" section), which correlates strongly with developmental progression in a large number of scRNA-seq datasets, including of the *Arabidopsis* root[2,13]. We benchmarked the developmental progression score directly against data from endodermis cells of specific developmental stages, and found a strong relationship between loss in transcriptional diversity and endodermis developmental stage (Supplementary Fig. 5B)[2]. For each cell of the endodermal sub-types, we calculated the average developmental

progression of its 25 nearest neighbors among root scRNA-seq cells (Supplementary Fig. 5C, D) and found, assigning this average to each scATAC endodermis cell, a trend of developmental progression among the endodermis sub-types (Fig. 3c). This result was robust to changes in the number of neighbors used to identify similar cells from scRNA-seq data (Supplementary Fig. 5E). We confirmed the orientation of the developmental trajectory among these clusters using (1) signature scores computed for early, middle, and late stage-associated genes of the endodermis[3]; and (2) correlation to bulk transcriptomes from FACS-based isolation of endodermis cells at different developmental stages (Supplementary Fig. 5F, G)[1]. This trend was the same if we calculated the developmental progression metric based on scATAC-seq data alone (Supplementary Fig. 5H), though the correlation to the transcriptional metric was weak overall (Supplementary Fig. 5I)[13]. Cells from sub-type 1 were the least

developed, while cells from sub-type 3 tended to co-occur with the most mature endodermal cells in the co-embedded graph (Fig. 3c). We conclude that the three endodermal sub-types primarily represent cells of differing developmental progression, and that differences in chromatin accessibility are able to capture this stratification of endodermis maturity.

Developmental progression in the root is associated with increased ploidy through endoreduplication[19]. Using the transcriptional-based metric for endoreduplication described above, we examined the predicted ploidy of orthogonally-classified cells derived from scRNA-seq (Supplementary Fig. 5J) and from the nearest RNA-seq neighbors of each endodermis sub-type (Supplementary Fig. 5K). We found that the younger endodermis sub-type 1 cells had mostly 2n neighbor cells, while the more mature sub-types 2 and 3 had mostly endoreduplicated neighbor cells, with similar levels in each (Fig. 3d).

To better understand the differing transcriptional and chromatin accessibility patterns among endodermis sub-types, we analyzed differentially expressed genes from each endodermis sub-type. The early endodermis type, which is not yet endoreduplicated, showed an enrichment of genes (Supplementary Data 4) involved in Casparian strip formation (CASP3, CASP5) and wax biosynthesis (HHT1). The intermediate sub-type 2 also showed enrichment for genes involved in Casparian strip formation (CASP3, CASP4, CASP5, and GSO1), as well as mechanosensitive ion channels (MSL4, MSL6, and MSL10) (Supplementary Data 5). The most advanced endodermis sub-type 3 showed enrichment for stress responses and metabolism of toxic compounds, kinase activity, and aquaporin water channels (Supplementary Data 6), consistent with this mature endodermis cell type modulating water permeability via aquaporins as well as through suberization[34]. We also identified putative regulators of these stages by looking for transcription factors among the genes that showed specificity for each endodermis cluster. The earlier endodermis type showed a single upregulated transcription factor, ERF54, while the intermediate sub-type showed 14 upregulated transcription factors, including KNAT7, SOMNUS, and HAT22. MYB36, which was found expressed in the later endodermis type, activates genes involved in Casparian strip formation and regulates a crucial transition toward differentiation in the endodermis[35]. Because MYB36 regulates early steps of endodermis differentiation[3,35], this result suggests that some more mature endodermis types may be absent in these data, perhaps due to technical differences in their ability to be lysed during nuclear extraction (see "Methods" section).

We used a list of known cell-cycle marker genes (https://www.arabidopsis.org/) to generate a signature score marking proliferating cells. This signature score identified cycling cells in other cell types, such as early epidermis cells near the quiescent center (Supplementary Fig. 6A, B) in a meta-analysis of previously published scRNA-seq data. However, when this signature score was transferred to the scATAC-seq endodermis clusters by the nearest neighbor procedure described in Supplementary Fig. 3C, we observed no differences corresponding to each endodermis sub-type (Supplementary Fig. 6C). We conclude that cell cycle does not distinguish the endodermis sub-types.

Overall, the combined information gained from transcriptional signatures of developmental progression and endoreduplication highlights the importance of integrating both open chromatin and transcriptional profiling, to identify cell types or cell states that may have otherwise been obscured in a single data type.

**Predicting regulatory events using integrated scRNA and scATAC data.** We previously identified transcription factor binding motifs that were enriched at cell type-specific peaks in the

root (Fig. 1d). While individual motifs may be associated with binding and activation by transcription factors, a sequence-level analysis cannot distinguish among the many members of plant transcription factor families that share near-identical sequence preferences. For example, WRKY family motifs were highly enriched among epidermis and cortex accessible sites, but this family contains >50 individual genes. In order to narrow down this list of genes to a few possible candidates, we leveraged our nearest-neighbor annotation approach (Supplementary Fig. 3C) to examine expression levels of all WRKY family transcription factors in the scATAC data (Fig. 4a). Overall, we found that the majority of WRKY members showed expression in the epidermis, cortex, or epidermal precursor cells (Fig. 4a), though some members showed stele-specific expression. To identify the most likely members to bind the abundance of motifs in epidermis-specific peaks, we ranked these genes by their specificity in the epidermis. The top four most epidermis specific genes, WRKY75, WRKY9, WRKY6, and TTG2 (Fig. 4a), have documented roles in root development[28,36–38]. TTG2 showed strong specificity for the epidermis, but we also predict expression in some cortex and precursor cells (Fig. 4b). Two key interacting factors of TTG2 that also contribute to epidermis development, GL2 and TTG1[39,40], showed epidermis expression and had correlated patterns (Pearson correlation with TTG2 across cells for GL2 = 0.91, and TTG1 = 0.47) across all cells (Supplementary Fig. 7A, B).

Given the important role of TTG2 in specification of atrichoblast fate in the epidermis, we examined the consequences of its expression on accessibility of individual peaks. Inference of individual regulatory events, particularly those involving transcription factors, has long been a goal of studies that profile accessibility at regulatory sites in bulk tissue. The varied cell states revealed by single-cell profiling data, even those within a cell type, allow higher-resolution inference of these events. To identify accessible sites that showed altered accessibility as a function of transcription factor expression, we used a linear regression approach. We identified 617 peaks that showed significant (q-value < 0.05) associations with TTG2 expression levels (Supplementary Data 7). To visualize these associations using scATAC data, we pseudo-bulked epidermis, cortex, and c/e precursor cells into four equal-sized bins based on their level of TTG2 expression (Fig. 4c). We observed peaks whose accessibility increases (Fig. 4c, top and lower-left panels) and decreases (Fig. 4c, lower-right panel) in cells with increasing levels TTG2 expression. Most significant associations were positive, such that increased TTG2 expression led to increased peak accessibility (Fig. 4d). Using DAP-seq data for TTG2, we examined whether peaks with either positive or negative associations contain TTG2 binding sites[27]. Positive associations occurred whether or not a WRKY binding motif was present in the associated peak (Fig. 4c, d), suggesting that the role of WRKY transcription factors in specification of the epidermis likely requires both direct and indirect regulatory events. Of peaks with significant (q-value < 0.05) positive associations with TTG2 expression, 80% of these contained a WRKY binding motif, while only 38% of the peaks with negative associations contained a binding motif (Fig. 4d). Overall, this analysis identifies transcription factors and putative target sites that constitute regulatory events important for specifying cell types; these genes and regulatory sites are good candidates for further functional studies.

## Discussion
By profiling chromatin accessibility in the A. thaliana root at single-cell resolution, we assessed cell types, developmental stages, the transcription factors likely driving these stages and DNA copy number changes. We assigned over 5000 root cells to tissues and

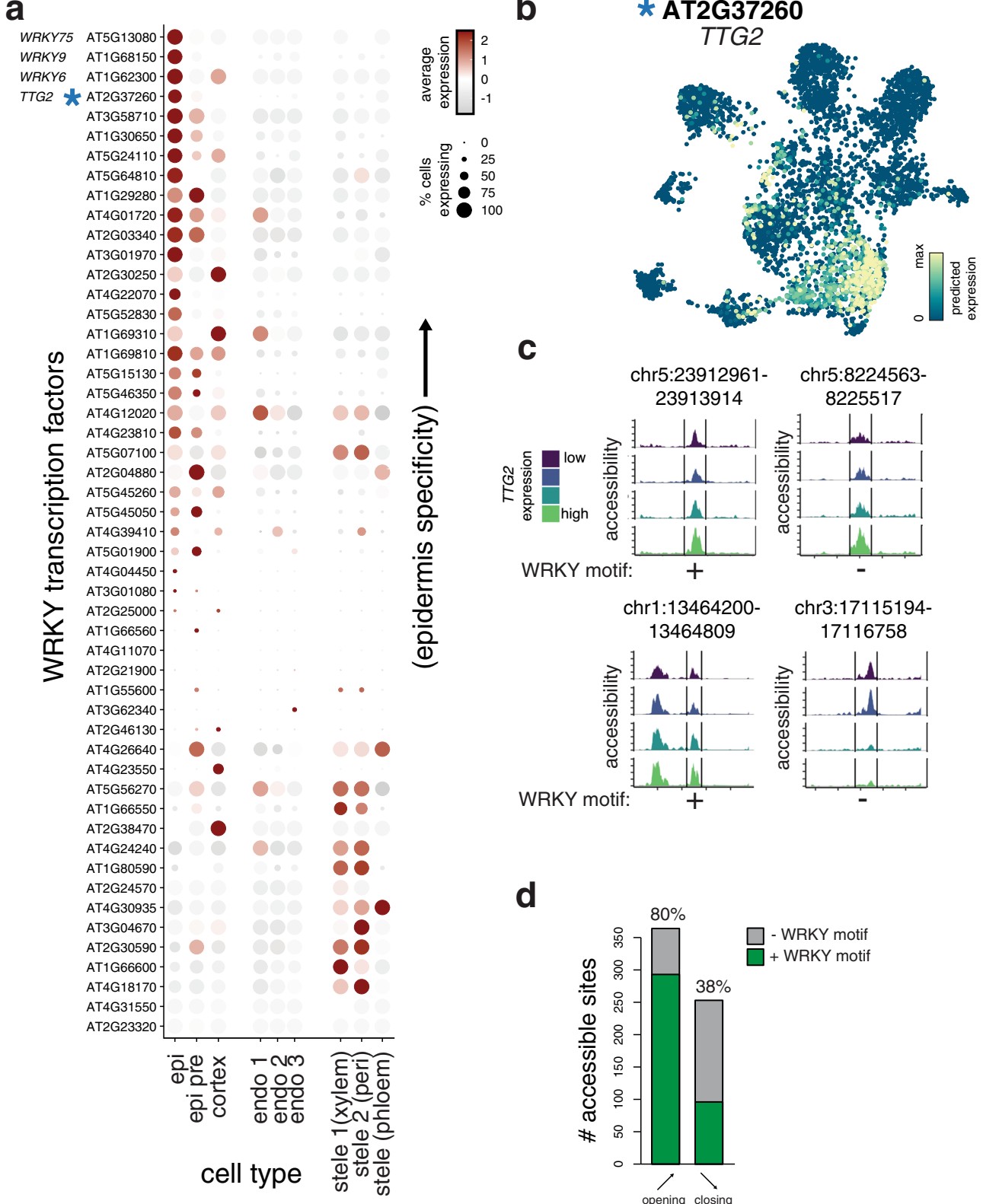

cell types, demonstrating that these assignments are concordant with single-cell transcriptomic studies. These results answer an unresolved question in plant gene regulation: does the paucity of dynamic open chromatin sites seen in bulk profiling experiments represent an accurate reflection of uniform gene regulation in *A. thaliana* or does it reflect a confounding effect of bulk studies? We found that distinct root cell types show unique patterns of

open chromatin sites, with approximately 1/3 of all accessible sites showing cell type-specific patterns. This estimate greatly exceeds the earlier estimates from bulk studies of only 5-10% of accessible sites showing tissue-specificity or condition-specificity[9], presumably due in part to tissue heterogeneity.

Although this single-cell ATAC study discovered many more dynamic accessible sites, the correlation between dynamic

**Fig. 4 Prediction of candidate regulatory transcription factors from integrated scATAC and scRNA data. a** Dotplot heatmap showing predicted expression of all WRKY family transcription factors across all cells. The color of each dot indicates the magnitude of predicted expression of each gene and the size of each dot represents the fraction of cells in each cell type showing expression at that gene; genes (rows) are ordered by the specificity of their epidermis expression. **b** UMAP plot of cells derived from scATAC experiment, but colored by predicted expression of an epidermis-specific WRKY transcription factor, *TTG2*. **c** Pseudo-bulked accessibility tracks of epidermis peaks whose accessibility showed a significant association with predicted *TTG2* expression. Cells with higher *TTG2* expression are shown in lighter shades. All panels show examples of significant ($q < 0.05$) positive associations of *TTG2* expression with peak accessibility, with exception of the lower right panel. The presence or absence of a WRKY binding motif is indicated below each peak. **d** Barplot showing fraction of WRKY binding motifs in peaks of the epidermis, cortex, and pre-cursor type that showed significant association with *TTG2* expression. Peaks whose accessibility showed positive associations with expression are labeled as "opening"; those with negative associations are labeled as "closing".

accessibility and gene expression in single cells remained poor, reminiscent of the equally poor correlation seen in bulk studies. These data types would be integrated more faithfully in a true co-assay experiment[25,41]. Furthermore, decisions made in relating transposition events to genes complicate analysis; by opting for a simplistic grouping of proximal promoter cuts with gene body cuts, rather than a more nuanced method[42], we may have introduced noise into the gene-level counts. Technical differences in nuclei versus cell-based assays, size selection, developmental stage, and sequencing depth may also contribute to differences between scRNA and scATAC datasets. While increasing the depth of our ATAC signal per cell may alleviate some of this noise, we argue that the poor correlation between chromatin accessibility and gene expression is not a function of data quality. Instead, we propose that this weak correlation reflects the complex nature of regulatory processes underlying development, and the differential aspects of regulation captured in scATAC-seq and scRNA-seq data, which were notably divergent in the scATAC-specific endodermis sub-types.

We found three distinct endodermis cell clusters that differed in transcriptional complexity and endoreduplication, consistent with the clusters representing early, middle, and late stages of endodermis development previously characterized with scRNA-seq[3]. However, at the level of individual marker genes, like the early-expressed *MYB36*, a more complex scenario emerged: accessibility of *MYB36* was similar across the three endodermis subtypes, and predicted expression of *MYB36* transcript, while present in all three subtypes, was highest in "endodermis 3", the cluster with the most advanced developmental progression and highest endoreduplication. Without a direct, one-to-one relationship between accessibility and expression at individual loci, we do not expect a one-to-one match for every endodermis scATAC cell cluster with one found in scRNA space.

Although the correlation of chromatin accessibility and gene expression is weak at the level of individual loci, either the entirety of a cell's regulatory landscape or its transcriptome can independently capture its cell identity. It is this feature that allows joint co-embedding of both data types and the use of scRNA-seq data to annotate scATAC cells. Thus, while the patterns of both chromatin accessibility and gene expression contain information on cell identity and development, the relationships between these patterns are not well-ordered or parsimonious. For the many cells belonging to a distinct cell type, gene expression results from direct and indirect regulatory events involving tens or hundreds of transcription factors and chromatin remodelers that do not necessarily act in concert. For any individual locus, then, the expectation that average accessibility predicts average expression breaks down. Without a simple one-to-one model to explain regulatory output, we are left with significant heterogeneity within and between cell types, and a subset of convergent expression or accessibility patterns that define cell type specificity. Alternative explanations for the discrepancy in accessibility and expression include: (1) maintenance of cell identity requires that a

cell's accessibility and expression profile stably reflect the convergent pattern for that cell type only a fraction of the time and/or; (2) cells have multiple accessibility and expression patterns that are sufficient to maintain cell identity and together constitute the convergent patterns we observe. In both scenarios, the heterogeneity in cell type specification will be buffered by factors outside chromatin accessibility or gene expression, such as spatial location in tissue, metabolic determinants of cell function, or developmental age.

We posit that scATAC-seq data combined with scRNA-seq data will ultimately resolve these alternatives by enabling mechanistic models of gene regulatory networks. scATAC-seq data alone are sufficient to identify the full set of accessible sites in the *Arabidopsis* genome, and examination of the transcription factor motifs within these sites can enable predictions of regulatory networks. However, many plant transcription factor families are large, some containing over fifty members that recognize near identical motifs. Thus, the accessibility data must be integrated with single-cell expression data that capture cell type-specific expression of transcription factors in order to narrow down the most probable transcription factors that are enacting individual regulatory events. The simple regression framework provided in this work is only a small step toward more complicated models that capture other relevant sources of heterogeneity. Building higher resolution models of key regulatory events will require the expression level of individual transcription factors in a cell type, the accessibility of individual peaks in this cell type and the presence of binding motifs corresponding to the relevant transcription factors. Theoretically, a comprehensive capture of cell states with both open chromatin and transcriptional profiling will allow the ordering of gene regulatory events, and the larger scale ordering of regulatory programs that underlie development. The ability to take single-cell measurements over distinct developmental stages will also increase the sampling of key regulatory events. Ultimately, achieving the goal of building models of gene regulatory events underlying development will require ever larger datasets to fully capture the range of possible cell states.

In the future, single-cell studies of more complex plant tissues in crops and other species will necessitate larger numbers of profiled cells and higher numbers of cuts per cell. Two recent studies have begun to explore chromatin accessibility in crops in single-cells[43,44]. Deeper coverage in future datasets should enhance our ability to detect rare cell types and more confidently predict copy number from accessibility data alone. In this way, approaches that maximize the number of cells profiled at low cost, such as single-cell combinatorial indexing[45], will be critical. Annotation in future studies will also present a substantial challenge if a rich literature and genomic analyses, including single-cell transcriptome profiles, are not available. Nevertheless, as shown in this proof-of-principle study of the well-characterized *A. thaliana* root, the knowledge gained should eventually allow us to manipulate gene expression and organismal phenotype in a targeted manner.

## Methods

**Plant materials**. Genotype: *Arabidopsis thaliana* ecotype Col-0 INTACT line *UBQ10*:NTF::*ACT2*:BirA (available from ABRC, stock CS68649). Growth conditions: LD (16 h light/8 h dark), 22 °C, ~100 µmol m2s, 50% RH. Sample: whole roots, harvested 12 days after germination, from seedlings grown vertically on MS + 1% sucrose, atop filter paper (to facilitate root harvesting).

**Nuclei isolation and scATAC-seq**. Nuclei were isolated following a modified version of the protocol described in Giuliano et al. as follows: 1 g of roots was split in two batches of 0.5 g, and each batch chopped with a razor blade in 1 mL of Buffer A (0.8 M sucrose, 10 mM MgCl2, 25 mM Tris-HCl pH 8.0 and 1× Protease Inhibitor Tablet)[46]. Extracts were combined, final volume increased to 5 mL with Buffer A, and incubated on ice for 10 min, with gentle swirling. The combined extract was filtered through miracloth, passed through a 26ga syringe five times and re-filtered through a 40 µm cell strainer (BD Falcon). After centrifugation at 2000 × g 5 min, the pellet was resuspended in 1 mL Buffer B (0.4 M sucrose, 10 mM MgCl2, 25 mM Tris-HCl pH 8.0, 1× Protease Inhibitor Tablet, 1% Triton X − 100) and loaded atop a 2-step 25/75 Percoll gradient (1 volume 25% Percoll in Buffer B over 1 volume 75% Percoll in Buffer B). After centrifugation at 2500 × g for 15 min, nuclei were collected either at the 25/75 interface or in the subjacent 75 fraction, washed with 5 vols of Buffer B and recovered by centrifugation at 1700 × g for 5 min. The nuclei pellet was resuspended in 100 µL Buffer B + 1% BSA and any nuclei clumps broken down by pipetting up and down multiple times. Nuclei yield with this protocol was ~94,000 nuclei per gram of roots (fresh weight).

scATAC-seq libraries were built using the 10× Genomics Chromium Single Cell ATAC Solution platform, following manufacturer's recommendations. Before transposition, nuclei were spun 5 min at 1500 × g and resuspended in 10× Genomics Diluted Nuclei Buffer, at a concentration of 3200 nuclei/µL. Five microliter of nuclei suspension were used for transposition (16,000 nuclei being the maximum input recommended for 10× Chromium, and 10,000 nuclei being the expected recovery).

**Combining and processing of root scRNA-seq data**. Samples were processed using the CellRanger v1.2.0 pipeline from 10× Genomics, including updated filtering of "halflet" cells that emerge due to multiply-barcoded droplets.

**Integration of scRNA and scATAC data**. The R package Seurat version 3.1.5 was used to align and co-embed the scATAC-seq data with scRNA-seq data published by Ryu et al.[5], and to transfer cell type labels from the scRNA data to the scATAC data[30,47].

The standard workflow and default parameters as described in the Seurat vignette "PBMC scATAC-seq Vignette" (https://satijalab.org/seurat/archive/v3.1/atacseq_integration_vignette) were used with the exception that all features (genes) were used when identifying transfer anchors, and performing the co-embedding rather than a set of "variable" features as used in the vignette. Briefly this workflow is as follows:

An anchor set was established with the function "FindTransferAnchors()" linking the two datasets. Cell type annotations were transferred from the scRNA-seq data to the scATAC data using the function "TransferData()". Imputed RNA-seq count data was generated for the scATAC cells, again using the "TransferData ()" function. The imputed RNA data was then merged with the true scRNA-seq dataset and embedded in 2D UMAP space using "Seurat" functions[29].

A co-embedding was performed with a super-set of published scRNA-seq data[2,3,5]. In the co-embedded space the scATAC-seq were found to be most closely co-located with data from root tips[5]. Based on this observation co-embedding was performed with solely with root tip dataset[5].

**Nearest neighbor analysis for transcriptional characterization of cells identified in scATAC assay**. To annotate cells from the scATAC-seq assay with transcriptional features, we used average feature values from the nearest RNA neighbors in our co-embedded data (Fig. 2a). In short, the "distances" package in R was used to extract cell labels for the 25 nearest neighbors of each scATAC cell. For a feature of interest (individual gene expression, cell-cycle signature score, endoreduplication signature score, and developmental progression signature), we calculated the mean expression from the 25 scRNA cells, and assigned that mean score to each ATAC cell (Supplementary Fig. 3C).

**Endoreduplication signatures**. We identified endoreduplicated cells using two different approaches, the first using scRNA data, and the second using scATAC data. In the first approach (as in Fig. 3d, Supplementary Fig. 4D, E, and Supplementary Fig. 5B, J), validated sets of endoreduplication markers for 2N, 4N, and 8N cells were used to identify endoreduplicated cells in the scRNA data[19]. For these markers, we used the top 250 ranked genes ranked for each ploidy level. The signature scores were computed on normalized expression values for each gene using the monocle3 function "aggregate_gene_expression", producing a log-scale signature of ploidy. We used the nearest neighbor approach described above to transfer this transcriptional signature to scATAC cells. The average expression of each gene group was computed for each individual cell, and subsequently averaged per cluster to generate cell type-specific levels of each ploidy signature. To identify

clusters that were more likely to be endoreduplicated, rather than typical diploid cells, we examined, for each cluster, the ratio of the endoreduplicated signatures (4N or 8N) relative to the diploid (2N) signature. Clusters with a higher ratio are more likely to represent endoreduplicated cells. In the second approach (as in Supplementary Fig. 4A–C), the number of transposition events derived from scATAC data were used directly to identify endoreduplicated cells. We assumed that cells containing higher than average cuts per peak were more likely to be endoreduplicated, as the cut counts for a single peak in a diploid cell should rarely be above two. A peak with a cut count >2 may indicate an extra copy of the locus present in that cell. To identify cells more likely to be endoreduplicated, then, we examined the distribution of cuts per peak for all cells, but found this metric was strongly correlated with total UMIs per cell. To account for contribution total UMIs per cell, we used the relationship between the cuts/feature and total UMIs per cell to compute a Loess model fit (Supplementary Fig. 4B). We then used residuals of this model as a metric to identify cells that have higher cuts/feature than would be expected based on their total UMIs. We set an arbitrary threshold of >1 SD in the distribution of each cell's deviation from the fit line, and defined endoreduplicated cells as those beyond the threshold (Supplementary Fig. 4B). For each cell, a binary designation of endoreduplication was applied based on whether the cell crossed this threshold.

**Metrics for developmental progression**. Using the general premise that the number of unique genes expressed (transcriptional complexity) tends to be reduced across the developmental trajectory of a cell type as it moves from earlier to later stages[13], we devised a metric to approximate relative differences in developmental progression among cells. Measuring the number of unique genes expressed is distinct from measuring the number of UMIs or transcripts captured per cell, which can vary across cell types. To account for differential recovery of UMIs across cells in the transcriptional complexity measure, we modeled as a Loess fit the relationship between total UMIs captured and the number of unique genes expressed per cell. With this fit, we identified cells that have many more or fewer unique genes expressed than would be expected for cells over a range of captured UMIs. Developmental progression for each cell was defined as the residual of each point in this fit, allowing separation of earlier cells (more unique genes expressed than would be expected for a given number of captured UMIs) from later cells (fewer unique genes expressed than would be expected for a given number of captured UMIs).

A similar analysis as the above was conducted using the scATAC data alone[13]. By using the gene activity score matrix (summarizing all transposition events within each gene, and within 400 bp of its start site), we computed for each cell the total UMIs normalized by the total number of accessible genes (count ≥ 1). Much like the transcriptional complexity metric above, we expect this ratio to increase with developmental progression. As the total number of accessible genes decreases with developmental time, a larger fraction of UMIs are found in this smaller set of genes.

**Motif analysis**. Position weight matrices from the comprehensive DAP-seq dataset[27] were used as input into FIMO (v5.0.0)[48] to search for significant matches for each individual TF motif (adjusted *p*-value threshold <1e−5) in each of the scATAC peaks. With the output of this motif scan, we generated a matrix that tallied counts of each individual motif within each peak. Each individual motif in the DAP-seq dataset[27] has an associated TF family, and the counts per peak were averaged by family. To identify motifs whose counts were significantly associated with cell type-specific accessibility, we first generated, for each peak, a relative accessibility score by taking the mean accessibility of that peak in each cell cluster relative to the overall accessibility of that peak in all clusters. Next, we used a linear regression framework within Monocle3[49] to identify individual motifs whose counts showed strong positive or negative correlations with the cell type-specific accessibility score in each cell cluster. The effect size of each motif's contribution to cell type-specific accessibility is given as the β of the linear regression, shown as a mean across all transcription factors in the same family.

**Reporting summary**. Further information on research design is available in the Nature Research Reporting Summary linked to this article.

## Data availability

Data supporting the findings of this work are available within the paper and its Supplementary Information files. A reporting summary for this Article is available as a Supplementary Information file. The dataset generated and analyzed in the current study is publicly available; raw and processed data, including an R object containing all accessibility and predicted expression data for each cell, have been uploaded to GEO under accession GSE173834. Source data underlying Fig. 1a as well as cell annotations are provided as a Source Data file. Source data are provided with this paper.

## Code availability

We have provided R markdown files [https://github.com/mwdorrity/scatac_root] with code blocks sufficient to complete the primary processing of the data, generation of scATAC and scRNA co-embedding, analysis of motifs, and identification of transcription-factor mediated regulatory events.

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

## Acknowledgements

We thank Dr. Ken Jean-Baptiste and Dr. Kerry Bubb for valuable discussions on ATAC-seq analysis. We also thank Xavi Guitart for helpful discussions on endoreduplication. This work was supported by the National Science Foundation (RESEARCH-PGR grant 17488843) to S.F. and C.Q. This work was also supported by NIH grant 1RM1HG010461 to C.Q. and S.F and R01-GM079712 to C.Q. and J.C.

## Author contributions

M.W.D., C.M.A., C.Q., S.F., and J.C. conceived and interpreted experiments. M.W.D., C.Q., S.F., and J.C. wrote the manuscript. C.M.A., A.V., and J.C. collected single-cell data. M.W.D. conducted data analysis and M.H. assisted with dataset integration.

## Competing interests

The authors declare no competing interests.
