## [Peer Review File · Nature Communications]

REVIEWER COMMENTS

Reviewer #1 (Remarks to the Author):

In "The regulatory landscape of Arabidopsis thaliana roots at single-cell resolution", the authors generate a snATAC-seq library for Arabidopsis root and then perform a series of correlative/bioinformatic analyses to compare snATAC to scRNA-seq and identify insights into gene regulation, endoreduplication, etc. during root development. The primary findings of the paper are 1) snATAC-seq, like scRNA-seq, can be used to uniquely profile (most but not all) root cell types; 2) endoreduplication can (potentially) be identified using snATAC-seq data; 3) snATAC-seq and scRNA-seq results are consistent with one another; 4) snATAC-seq identified 3 distinct populations of endodermis cells that seem to be from different developmental stages; 4) snATAC can be used to identify enriched motifs and scRNA-seq can be used to narrow down the list of candidate TFs that may act on the motifs. snATAC-seq on plant tissue has not, to my knowledge, been previously described in the literature, so there is some novelty to the data generated. While plant tissue snATAC-seq and combining snATAC and scRNA-seq are interesting technical advancements, the data generated (a single snATAC-seq library on one tissue) are very limited. And the scientific findings gleaned from the data are largely correlative and require independent experimental validation. Without some independent experimental validation to support some of the claims, this manuscript does not seem appropriate for Nature Communications.

Major

- 1) Line 119-121: "Cell type-specific ATAC tracks that resemble those obtained in prior whole tissue and cell type enrichment-based ATAC-seq studies for the root (Figure 1B)." First, should this be Fig. S1B? Second, the statement in reference to cell type enrichment based ATAC-seq doesn't seem to be supported by any data shown. Overall, the first section would benefit from more comparisons to previous bulk and INTACT-enriched (cell type specific) ATAC-seq data from Arabidopsis root to demonstrate the unique findings from the snATAC data and validate the cell cluster classifications. What percentage of peaks are found in snATAC that were not found by previous bulk data (and vice versa)? How well do the snATAC clusters correlate with data from INTACT-enriched ATAC-seq (if such exists for root cell types)?
- 2) Paragraph starting line 123: I'm confused about how exactly the scRNA-seq marker genes were used to assign identities to snATAC-seq clusters if the "ATAC maker genes did not show a strong overlap with RNA-based marker genes." Exactly how poor was the overlap? I.e., what % of snATAC-seq marker genes were scRNA-seq marker genes and vice versa? How well are known marker genes (from traditional, rather than single cell methods) captured by snATAC?
- 3) I have substantial concerns about the endoreduplication findings throughout the paper. There is no description in the Methods to describe any of these analyses, and I'm not entirely clear what was done at various points in the paper to establish endoreplication since there seem to be multiple definitions used at various points (e.g., snATAC-seq cuts/feature, "transcriptional signatures of endoreduplication"). More specifically, the section beginning at Line 183 may well be describing a technical artifact of snATAC-seq since it's not clear what the empirical range of cuts/feature would be using snATAC-seq on a cell population with an expected genome $n=2$. How was the endoreplication $\log_{10}(\text{cuts/feature})$ threshold shown in Figure S5B selected? It looks completely arbitrary since there aren't any multimodal features in the plot. I don't have as much hands-on experience with snATAC, but with scRNA-seq data, my own experience is that within the same library there are wide ranges of UMIs captured per cell, and cells w/ larger UMI counts cluster much more cleanly into clear cut cell types than those with fewer UMI counts. This could be evidence of endoreplication, or it could just be a technical artifact of varying degrees of molecule capture that leads to better clustering of some cells because of the overall sparsity of the data. Without independent (not single cell) verification or an understanding of what the UMI or cuts/feature distribution is in truly diploid cells, it's not clear to me which is true.
- 4) Is there a way to independently verify the 3 endodermis clusters since some (e.g., endo 1) were

only found in the snATAC-seq data and the expression of the genes the cluster-specific peaks presumably regulate isn't correlated with cluster?

5) Paragraph starting at Line 321: is the developmental trajectory of the clusters or gene expression backwards here? Myb36 is described as being expressed in the late endodermis cluster, but previous studies have shown that it regulates the expression of Casparian strip genes and is most strongly expressed in the elongation zone (PMID: 26371322). And the gene expression described here is seemingly in the opposite developmental direction of previous scRNA-seq analysis of endodermis expression (e.g., PMID: 31091459).

6) The last section (related to Figure 4) was the most interesting avenue of the paper, but, it too, is entirely correlative. At the very least, is there existing TTG2 ChIP-seq from bulk root or DAP-seq data to support the approach and demonstrate that the identified sites are bona fide TTG2 binding sites?

7) The manuscript overall looks like it was rather hastily put together, which made it unnecessarily confusing in a lot of places. The supplementary figures are cited wildly out of order. There are places where acronyms or abbreviations aren't clarified (e.g., c/e pre should be defined in Fig. 1 legend). Figure titles on the figure pages in a few places don't match the titles on the figure legend pages. There are no Methods to describe the endoreduplication analyses. Line 505: software version missing. Data were not made available to reviewers for independent analysis. Most of these individually are minor issues, but collectively they make reading and reviewing the paper extremely challenging. Please take some time to check the overall organization of the paper.

Minor

1) Section starting Line 98: what percentage of peaks from snATAC are proximal versus distal? Was either class (proximal or distal) enriched among those that are cell type specific?

2) Line 134: is the inability to elucidate the identities of 3 clusters a snATAC-seq sensitivity issue? Do these have overall lower peaks numbers/UMIs than other clusters? Would sequencing the library deeper rescue these from obscurity and/or help deconvolve hair and non-hair epidermis?

3) Line 231-232: should this be cluster endo #1, rather than #3?

4) Line 257: is there a paper citation supporting BLUEJAY's role in endodermis differentiation?

5) One of the reasons this paper is challenging to follow and read is that the t-SNE/UMAP embeddings can change abruptly throughout the text, making it impossible to compare related information between different figures. Or different UMAPs are being used to support statements that they probably shouldn't be used to support. For example, Figure 3 shows one version of the snATAC-seq embedding with all cells or just zoomed to endodermis. Figure S3C shows a different embedding combining snATAC and scRNA-seq for endodermis cells. For the right half of Figure S3C (the gene expression plots), this would be clearer to understand in relation to the 3 endo clusters if nearest neighbor expression was shown in the version of the UMAP/t-SNE shown in Figure 3. Figure S4B shows yet another UMAP version to justify the lack of difference between cell cycle state between the 3 endo clusters, but the 3 endo clusters aren't even distinguishable on this version of the plot because they aren't seen in the scRNA-seq data. How can this actually support this statement?

6) Line 263: what is it then that distinguishes the clusters? Are these peaks possibly marking some other noncoding sequence function other than positive (promoter/enhancer) regulation?

7) Paragraph Line 276: is there a series of typos in this paragraph, or is it really a paragraph about epidermis in a section otherwise devoted to endodermis? If it is about endodermis, the result is completely expected, isn't it, given that everything except the precursor cells should no longer be dividing?

8) Line 295-296: the plot shown in Figure 3C is completely different than Figure S3F. How exactly are these trends the same?

9) Paragraph p303: The information in this paragraph pertaining to non-endodermis cell types would be better suited for the earlier discussion of endoreduplication.

Reviewer #2 (Remarks to the Author):

The authors present an interesting study combining new scATAC-seq data and published scRNA-seq data on the *Arabidopsis thaliana* root to investigate whether scATAC-seq can be used to differentiate cell identities, and whether the combination of single-cell RNA-seq and ATAC-seq data enhances the resolution of the cell regulatory landscape in roots. To my knowledge, this is the first scATAC-seq study in plants. Interesting results include the observation that chromatin patterns are more cell type-specific than previously thought, that scATAC-seq can distinguish cell types and in some instances developmental stages of cell types, and that combination of scATAC-seq and scRNA-seq data can resolve regulatory processes influencing cell type specification (with a case study on TTG2). Overall, I think this is an interesting and novel study, but I do have a few major concerns to be addressed.

Major comments

- The methods section of the manuscript needs to be amended substantially. For many of the computational analyses, there are no methods entries and the description in the results section is not sufficient to understand what was done exactly. For instance, the authors use a cutoff on the mean number of ATAC cuts per peak across cells to classify cells as endoreduplicated or not, but it is unclear what criterion this cutoff was based on. Similarly, it is unclear how the $8n/2n$ and $4n/2n$ scRNA-seq measures for endoreduplication were calculated, which genes were involved in these calculations, or why these ratios were chosen and not the $16n/2n$ ratio or the $16n/4n$ ratio. Which ratios give information about which endoreduplication levels? Other methods statements in the results that do not give enough information on what was done include line 271-274 'we added annotations for cell cycle, developmental progression and endoreduplication to the combined root scRNA-seq data and used data integration (as in Figure 2) ...', line 276-277 'we used a list of known cell-cycle marker genes to generate a signature score marking proliferating cells (*Arabidopsis.org*).', line 283-285 'We assessed developmental progression with two orthogonal methods: (i) correlation with published bulk expression data taken along longitudinal sections of the root and (ii) a modified measure of loss in transcriptional diversity (see Methods)' (for which there is no methods entry), and others. On line 156-157, the authors mention that they used a set of representative motifs for all *A. thaliana* TF gene families and nearly every individual transcription factor, but I don't see any analyses involving motifs for individual TFs. It is not clear how many and which motifs exactly were included in the analysis.
- Interpretation of the results is complicated by the fact that the scATAC-seq data and scRNA-seq data were not generated on the same cells in the same experiment (a study in mammalia where scATAC-seq and scRNA-seq are done simultaneously came out recently : <https://doi.org/10.1101/gr.257840.119>). An additional complication is that the authors use whole roots for scATAC-seq while the RNA-seq data they compare to (Ryu et al. 2019 <https://doi.org/10.1104/pp.18.01482>) was generated on root tips. In other words, most of the cells profiled by ATAC-seq should be older than those profiled in the scRNA-seq dataset. This may explain to some extent the lack of correspondence in e.g. marker genes between both datasets. Definitive answers on the correspondence between scATAC-seq and scRNA-seq data in plants will have to wait until both are generated on the same cells of the same samples. Another possible reason for the observed relative lack of correspondence between chromatin accessibility and gene expression might be the inherently noisy nature of single-cell datasets, not only due to cellular stochasticity but also due to technical effects (low coverage per cell). The authors should comment on this in the discussion.
- on line 185-186, the authors state that 'in contrast to scRNA-seq data, scATAC-seq data can provide insight into DNA copy number and its impact on gene regulation.' On the other hand, in the following sections they establish that scRNA-seq derived predictions of DNA copy number (endoreduplication levels) are in fact superior. This begs the question whether there is added value to using scATAC-seq for predicting endoreduplication levels of cells.
- The manuscript convincingly shows that the scATAC-seq data captures three distinct endodermis developmental stages. Could the authors elaborate on why such subclusters would only be detectable for endodermis and not for other tissues? The authors further state on line 234-236 that 'The three distinct ATAC clusters that were assigned an "endodermis" label with this approach are a striking

example of scATAC data yielding greater stratification of cell types than the generally richer scRNA data.' It is however worth noting that there are methods available to infer developmental trajectories within a cell type from scRNA-seq data, see e.g. <https://doi.org/10.1038/s41587-019-0071-9> .

Minor comments

- the ordering of manuscript sections and figures is sometimes confusing. The section on endoreduplication starting at line 183 for instance refers to 'orthogonal measurements of endoreplication', referring to scRNA-seq-based inferences that are only discussed several pages further on.
- on line 212-213, the authors state that the root tissue sampled for the scATAC-seq experiment and previous scRNA-seq experiments was similar, while the methods state that whole roots of 12 day-old seedlings were used, as opposed to root tips of 5 day-old seedlings in Ryu et al. 2019 (<https://doi.org/10.1104/pp.18.01482>).
- Some of the references, in particular in the methods section and figure legends, are not in the bibliography and are formatted differently, e.g. Jean-Baptiste et al. 2019. Some other references are missing, e.g. for the gene functions discussed in line 248-258.
- Supplementary figure 3 : panel D is not discussed in the legend. Also 'subtype 3' is mentioned underneath panel C but it is unclear why. In panels E and F, it is unclear whether the differences observed between the bar chart means are statistically significant. In panels H, I and J, color legends are missing.

Reviewer #3 (Remarks to the Author):

In this manuscript, Dorrity et al. report the measurement of single cell chromatin accessibility in Arabidopsis roots using ATAC-seq. Chromatin accessibility is the gatekeeper of transcription factor binding. This study is important since, as outlined by the authors, recent studies uncovered overall invariable chromatin accessibility in different plant organs, which could reflect cell type heterogeneity or important regulatory layer of plant transcription plasticity.

Analyzing 5283 single nuclei resolved nine distinct cell clusters. The identity of these cell populations was assigned combining TSS-proximal accessibility, reflecting gene activity, with scRNA-seq profiles. Similarly to the whole organ level, TSS-proximal sc-accessibility did not show high correlation with gene expression profiles. Nevertheless, combining scATAC-seq with scRNA-seq allowed focusing on transcription factors and their combinations that are unique to different cell types.

The authors identified three clusters of endodermal cells in the scATAC-seq data. The loss in transcriptional diversity in these clusters led to suggest that they were related to different developmental stages. The progression of developmental states in these clusters and also in epidermal cells was further supported by elevated endoreduplication.

This study is important for pioneering measurements of sc-chromatin accessibility in plants. On the scientific merit, the study illuminates the level of chromatin accessibility specificity in different cell types and apply high-end tools to analyze the data and uncover novel potential transcription factors that regulate the genetic programs in these cells.

The data quality seems to be high and well presented.

Minor points to be addressed:

1. The lack of correlation between chromatin accessibility at gene proximal loci and gene expression is expected. However higher correlation was recently shown between gene expression and distant organ-specific accessible loci (ATAC-seq studies from Deal and Hakim labs). Please test this possibility in your data.
2. scRNA-seq data (references 2,5,6 in the manuscript) uncovered 9 to 15 distinct cell clusters and nine distinct cell clusters were resolved in this study. Please consider revising the statement in lines 235, 236 accordingly. In addition, the cell identities of the clusters from both technologies were not

identical (i.e. the different endodermis clusters were not uncovered in scRNA-seq). Please discuss the possible reasons for the differences between the two technologies, in addition to the number of analyzed cells. It would be useful if the authors could provide a quantitative view of the possible effect of the number of analyzed cells on the expected number of cell types to be uncovered by scATAC-seq.

3. A graphical map of the different root cell types (such as Fig. 1G in PMID: 30718350) would help non-plant scientists navigating the study.
4. Please name the supplemental tables.

Reviewer #4 (Remarks to the Author):

This is a first study to employ single cell ATAC seq to understand the gene regulatory programs in the Arabidopsis root, an important model system for plant development.

The study is overall very straightforward in applying existing experimental and computational tools. There are a couple of concerns that may impact the observations and conclusions:

- Perhaps there are so few cell type-specific regions because the authors are summarizing ATAC signal in gene bodies, and in turn include only up to 400nt upstream. Previous reporter gene studies typically take much larger upstream regions to recapitulate transcription (3kb).

So, on the one side, this will likely introduce noise since the majority of gene body space does not represent regulatory elements with dynamic accessibility, while excluding most of the genome that is thought to harbor such elements. Why not summarise at the gene level by assigning each peak to its closest (downstream) gene? To support the gene body quantification, in addition to the regulatory region browser shots in fig1, the authors should make a similar panel with representative dynamic gene bodies. Also, the authors should show pseudo-bulk browser track comparisons to published tissue isolations from reference 11 referred to on line 121.

- the authors state on lines 295 and 296 that the atac-seq-based developmental progression score shows the same trends as the rna-seq-based developmental progression metric which, if true, would be an interesting observation. But examination of Supp FigS3F and S3G shows, if anything, a relative inverse trend (progressively lower scores for cell types 1, 2, 3 as opposed to progressively increasing for the rna metric), and an overall poor correlation. Perhaps this is due to a difference in the way the scores are derived, but there is not sufficient information in the methods to determine how each was calculated. Furthermore, on lines 311 and 312 the authors refer to S3F and S3G again, but what seems to be in the context of endoreduplication, which leaves the reader confused as to what is actually being plotted and compared to what. Overall, the presentation of the developmental progression metrics in the text needs to be clearer and the description in the methods of how each was calculated is insufficient. The statement about the utility of ATAC quantification for determining developmental progression likely needs to be revised unless otherwise clarified.

- the number of cells and reads per cell is quite on the low side for the current state of the art. A calculation how far from saturation this coverage is would provide a relevant point for discussion.

Author Summary

Major changes:

- New analyses that directly compare scATAC-seq epidermal cell data with INTACT ATAC-seq of root hair and root non-hair cell types have been added.
- New endoreduplication analysis has been added to account statistically for total UMI count differences across cells.
- A new Supplementary Figure 2 has been added to show the patterns of traditional marker genes using both scATAC and scRNA data sets; Supplemental Table 3 has been added to show marker genes for each cell type.
- Major revisions to the Methods section have been added to explain the developmental progression and endoreduplication metrics described in the manuscript.

Updated Figure panels:

To better reflect the structure of the manuscript, Supplementary Figures have been reordered as follows:

Original Figure S1 -> revised Figure S1
Original Figure S2 -> revised Figure S3
Original Figure S3 -> revised Figure S5
Original Figure S4 -> revised Figure S6
Original Figure S5 -> revised Figure S4
Original Figure S6 -> revised Figure S7

Revised Figure S2 is newly added.

Figure S1C, S1D. Boxplots showing results of new analysis comparing INTACT ATAC-seq of epidermal cell types alongside controls.

Figure S2A. Panel showing key root tissues and cell layers for reference.

Figure S2B-D. Panels showing patterns of known marker genes in accessibility and predicted gene expression data in scATAC cells.

Figure S4B. Shows new statistical approach for identifying endoreduplicated cells from scATAC data, accounting for differences in total UMI counts.

Figure S4C. Now represent fractions of endoreduplicated cells in each cell type using updated methodology.

Figure S4F. New boxplot showing relationship between two methods of identifying endoreduplicated cells relying on either scRNA-seq and scATAC-seq data (see Methods).

Figure S5A. Label updated to more clearly annotate displayed genes.

Figure S5B. Panel now shows statistical significance.

Figure S5C, S5D. Legends added to each panel.

Figure S5F. Y-axis inverted to ensure that both scATAC- and scRNA-based metrics of developmental progression are displayed in the same way.

Figure S5H. Panel now shows statistical significance.

Figure S5I. Legends added.

All figure legends have been updated to reflect these changes.

REVIEWER COMMENTS

Reviewer #1 (Remarks to the Author):

In “The regulatory landscape of Arabidopsis thaliana roots at single-cell resolution”, the authors generate a snATAC-seq library for Arabidopsis root and then perform a series of correlative/bioinformatic analyses to compare snATAC to scRNA-seq and identify insights into gene regulation, endoreduplication, etc. during root development. The primary findings of the paper are 1) snATAC-seq, like scRNA-seq, can be used to uniquely profile (most but not all) root cell types; 2) endoreduplication can (potentially) be identified using snATAC-seq data; 3) snATAC-seq and scRNA-seq results are consistent with one another; 4) snATAC-seq identified 3 distinct populations of endodermis cells that seem to be from different developmental stages; 4) snATAC can be used to identify enriched motifs and scRNA-seq can be used to narrow down the list of candidate TFs that may act on the motifs. snATAC-seq on plant tissue has not, to my knowledge, been previously described in the literature, so there is some novelty to the data generated. While plant tissue snATAC-seq and combining snATAC and scRNA-seq are interesting technical advancements, the data generated (a single snATAC-seq library on one tissue) are very limited. And the scientific findings gleaned from the data are largely correlative and require independent experimental validation. Without some independent experimental validation to support some of the claims, this manuscript does not seem appropriate for Nature Communications.

Major

1) Line 119-121: “Cell type-specific ATAC tracks that resemble those obtained in prior whole tissue and cell type enrichment-based ATAC-seq studies for the root (Figure 1B).” First, should this be Fig. S1B?

This sentence has been changed to properly reference Figure S1B.

Second, the statement in reference to cell type enrichment based ATAC-seq doesn't seem to be supported by any data shown. Overall, the first section would benefit from more comparisons to previous bulk and INTACT-enriched (cell type specific) ATAC-seq data from Arabidopsis root to demonstrate the unique findings from the snATAC data and validate the cell cluster classifications.

We agree that further benchmarking using existing data sets would strengthen our results. However, only a few of these data sets exist. We have added an analysis comparing our scATAC data to INTACT-enriched ATAC-seq data from root epidermal cells. Using peaks identified in the scATAC data, we examined cutcount values derived from bulk ATAC data from INTACT preparations of root hair and root non-hair epidermal cells, and a whole seedling control sample as background. Consistent with earlier INTACT ATAC-seq results, we found that many of these peaks showed cutcount enrichments specific to the hair or non-hair samples relative to the whole seedling background. We then grouped all the scATAC peaks by the relative fold change in either the INTACT hair or INTACT non-hair samples, and examined relative cutcounts derived from the scATAC data itself. In doing so, we see that peaks that showed higher enrichments in INTACT ATAC-seq of both epidermal cell types also showed higher cutcounts in epidermal cells identified in the scATAC analysis. Furthermore, if we instead examine cutcounts derived from scATAC stele cells as a control, we see no such enrichment in peaks with higher cutcounts in root hair and non-hair INTACT samples. This analysis has been added as additional panels Figure S1C and S1D (shown below).

Furthermore, we have added plots for several known marker genes, using the scATAC signal and the inferred transcriptional signal, both of which should help clarify the cell cluster classifications. Revised text in the Results section of manuscript referring to these plots is below:

“Several traditional marker genes were used to facilitate annotation of root cell types (Figure S2B-D), as were marker genes identified in previous scRNA-seq studies (Supplementary Table 3).”

What percentage of peaks are found in snATAC that were not found by previous bulk data (and vice versa)? How well do the snATAC clusters correlate with data from INTACT-enriched ATAC-seq (if such exists for root cell types)?

We have added text to the main results of the manuscript to highlight the number of new peaks identified with the scATAC data compared to bulk ATAC-seq data. We found 1794 peaks from the scATAC assay for which we observed zero cuts in a bulk ATAC-seq assay from Maher et al. 2018. The additional text:

“Furthermore, the scATAC assay detected 1794 peaks that not been observed at appreciable levels in bulk ATAC-seq.”

Related to the above, we have added boxplots to show correlations to previous INTACT-enriched datasets in root; see Figure S1C and S1D.

2) Paragraph starting line 123: I’m confused about how exactly the scRNA-seq marker genes were used to assign identities to snATAC-seq clusters if the “ATAC maker genes did not show a strong overlap with RNA-based marker genes.” Exactly how poor was the overlap? I.e., what % of snATAC-seq marker genes were scRNA-seq marker genes and vice versa? How well are known marker genes (from traditional, rather than single cell methods) captured by snATAC?

We agree that these numbers are important, and have now provided this information. The overlap of scRNA-seq based marker genes derived from the combined dataset in McFaline-Figueroa et al. 2020 is provided below. Among genes that marked specific cell types in scATAC data, an average of 12.75% of these overlapped with a marker gene in scRNA-seq data. Supplemental Table 3 has been added to identify the matching genes, along with their adjusted p-value, in both scATAC and scRNA data sets.

Epidermis (cluster 0) = 8/36 (22.2%)
Stele 1 (cluster 1) = 1/27 (3.7%)
Stele 2 (cluster 2) = 1/10 (10%)
Cortex (cluster 3) = 1/6 (16.7%)
Endodermis 3 (cluster 4) = 3/25 (12%)
NA (cluster 5) = 0/7 (0%)
NA (cluster 6) = 0/0
cortex/endodermis precursor (cluster 7) = 0/5 (0%)
Endodermis 1 (cluster 8) = 4/17 (23.6%)

NA (cluster 9) = 0/0

Endodermis 2 (cluster 10) = 4/24 (16.7 %)

Stele phloem (cluster 11) = 12/53 (22.6%)

Additionally, we have added Figure S2, which contains marker gene plots for traditional marker genes (*AGL42*, *APL*, *COBL9*, *GL2*, *MYB46*, *SCR*, *SUC2*, *WER*, *WOL*, *WOX5*) in the major cell types of the root, showing their specificity as a measure of the summed open chromatin in their promoters, as well as the RNA expression level inferred from the co-embedding procedure. These analyses should make the qualitative summary from the main text more precise, and the main text has been revised to reflect this:

“Several traditional marker genes were used to facilitate annotation of root cell types (**Figure S2B-D**), as were marker genes identified in previous scRNA-seq studies (**Supplementary Table 3**).“

3) I have substantial concerns about the endoreduplication findings throughout the paper. There is no description in the Methods to describe any of these analyses, and I'm not entirely clear what was done at various points in the paper to establish endoreduplication since there seem to be multiple definitions used at various points (e.g., snATAC-seq cuts/feature, “transcriptional signatures of endoreduplication”).

We agree that the two approaches used to identify endoreduplicated cells were poorly described in the manuscript. We have added detailed descriptions of the endoreduplication analyses to the Methods section:

“We identified endoreduplicated cells using two different approaches, the first using scRNA data, and the second using scATAC data. In the first approach (as in **Figure 3D**, **Figure S4D**, **S4E**, **Figure S5B**, **S5I**), validated sets of endoreduplication markers for 2N, 4N, and 8N cells were used to identify endoreduplicated cells in the scRNA data.¹⁹ We used the nearest neighbor approach described above to transfer this transcriptional signature to scATAC cells. The average expression of each gene group was computed for each individual cell, and subsequently averaged per cluster to generate cell type-specific levels of each ploidy signature. To identify clusters that were more likely to be endoreduplicated, rather than typical diploid cells, we examined, for each cluster, the ratio of the endoreduplicated signatures (4N or 8N) relative to the diploid (2N) signature. Clusters with a higher ratio are more likely to represent endoreduplicated cells. In the second approach (as in **Figure S4A-C**), the number of transposition events derived from scATAC data were used directly to identify endoreduplicated cells. We assumed that cells containing higher than average cuts per peak were more likely to be endoreduplicated, as the cut counts for a single peak in a diploid cell should rarely be above two. A peak with a cut count >2 may indicate an extra copy of the locus present in that cell. To identify cells more likely to be endoreduplicated, then, we examined the distribution of cuts per peak for all cells, but found this metric was strongly correlated with total UMIs per cell. To account for

contribution total UMIs per cell, we used the relationship between the cuts/feature and total UMIs per cell to compute a Loess model fit (**Figure S4B**). We then used residuals of this model as a metric to identify cells that have higher cuts/feature than would be expected based on their total UMIs. We set an arbitrary threshold of >1 SD in the distribution of each cell's deviation from the fit line, and defined endoreduplicated cells as those beyond the threshold (**Figure S4B**). For each cell, a binary designation of endoreduplication was applied based on whether the cell crossed this threshold.”

More specifically, the section beginning at Line 183 may well be describing a technical artifact of snATAC-seq since it's not clear what the empirical range of cuts/feature would be using snATAC-seq on a cell population with an expected genome $n=2$. How was the endoreplication $\log_{10}(\text{cuts/feature})$ threshold shown in Figure S5B selected?

We have more clearly defined the selection of this (indeed arbitrary) threshold in the Methods (see above comment).

It looks completely arbitrary since there aren't any multimodal features in the plot. I don't have as much hands-on experience with snATAC, but with scRNA-seq data, my own experience is that within the same library there are wide ranges of UMIs captured per cell, and cells w/ larger UMI counts cluster much more cleanly into clear cut cell types than those with fewer UMI counts. This could be evidence of endoreplication, or it could just be a technical artifact of varying degrees of molecule capture that leads to better clustering of some cells because of the overall sparsity of the data. Without independent (not single cell) verification or an understanding of what the UMI or cuts/feature distribution is in truly diploid cells, it's not clear to me which is true.

The reviewer's comment highlights a flaw of the original scATAC-based metric of endoreduplication. We have taken this comment to heart and have re-structured this analysis to account statistically for total UMIs per cell. New Methods text detailing this approach can be found in the above comment.

As an example to illustrate the correction made, the panels below show the model of the relationship between cuts per peak and total UMIs per cell. Cells with higher cuts per peak than expected for their total UMI count (>1 SD, colored in blue) were classified as endoreduplicated (left panel below, Figure S4B). In this formulation, the residuals of the model show no correlation with total UMIs (right panel below, which does not appear in final manuscript, but is shown here for further clarity), indicating that the higher cuts per peak in these cells are not a technical consequence of higher read depth.

As an estimation of the reviewer's suggestion to look at the distribution of cuts/feature in diploid cells, we have stratified this updated scATAC-based metric by varying levels of the transcriptional endoreduplication metric which has been used previously to distinguish diploid cells from those with higher ploidy (Bhosale et al. 2018). Cells classified as endoreduplicated using the scATAC data in this way tended to show higher levels of the transcriptional signature of endoreduplication (left panel below). In this way, we maintain the conclusion that scATAC data contain some information on endoreduplication, although even when accounting for total UMIs per cell these data still show a relatively weak correlation with signatures based on validated endoreduplication markers (left panel below is Figure S4F, and right panel, which does not appear in the final manuscript, expands bins beyond the binary classification):

However, it is nearly impossible to disentangle the number of UMIs and endoreduplication, because in addition to technical variation in recovery, increased DNA copy number should increase UMI recovery for endoreduplicated cells. Accordingly, we have limited our conclusions based on the scATAC-based endoreduplication metric and instead rely on the previously used and likely more reliable transcriptional metric. Nevertheless, highlighting the potential of scATAC data to identify endoreduplicated cells should encourage progress in future studies, as some of the technical variation in recovery may be mitigated.

4) Is there a way to independently verify the 3 endodermis clusters since some (e.g., endo 1) were only found in the snATAC-seq data and the expression of the genes the cluster-specific peaks presumably regulate isn't correlated with cluster?

Indeed, we have shown that the cluster-specific accessible genes do not correlate well with patterns of gene expression in these cells (**Figure S5A**). We have suggested that these scATAC-seq-specific clusters are more likely to represent stages of endodermis development or endoreduplication, rather than individual cell types. It is interesting that this result is not independently verified in the scRNA-seq data, highlighting the utility of gathering multiple data sources to characterize molecular states.

We have added text to the Discussion to highlight this feature:

“Instead, we propose that this weak correlation reflects the complex nature of regulatory processes underlying development, and the differential aspects of regulation captured in scATAC-seq and scRNA-seq data, which were notably divergent in the scATAC-specific endodermis sub-types.”

5) Paragraph starting at Line 321: is the developmental trajectory of the clusters or gene expression backwards here? Myb36 is described as being expressed in the late endodermis cluster, but previous studies have shown that it regulates the expression of Casparian strip genes and is most strongly expressed in the elongation zone (PMID: 26371322). And the gene expression described here is seemingly in the opposite developmental direction of previous scRNA-seq analysis of endodermis expression (e.g., PMID: 31091459).

We agree that it is indeed surprising that MYB36, whose expression defines the early stages of endodermis differentiation, seems to show highest levels in the latest of the three endodermis clusters we identified. We think that the timing of the three clusters identified in the scATAC data represents earlier endodermis stages overall. One possibility is that the nuclei extraction procedure itself enriches for earlier stages because older cells are more resistant to lysis. If this were the case, the most mature cells may not be well-represented in the data set. Consistent with this possibility, we note that qualitatively, the Seurat alignment worked best with an scRNA data set enriched for root tips, rather than whole root (Ryu et al. 2019). The alignment algorithm produced more confident label transfers with the root tip dataset rather than with a combined data set (McFaline-Figueroa et al. 2020).

We have added a sentence to the results to acknowledge this possibility:

“Because *MYB36* regulates early steps of endodermis differentiation,^{3,36} this result suggests that some more mature endodermis types may be absent in these data, perhaps due to technical differences in their ability to be lysed during nuclear extraction (see Methods).”

6) The last section (related to Figure 4) was the most interesting avenue of the paper, but, it too, is entirely correlative. At the very least, is there existing TTG2 ChIP-seq from bulk root or DAP-seq data to support the approach and demonstrate that the identified sites are bona fide TTG2 binding sites?

While we recognize that this section is indeed correlative, we think it represents a step forward in how to analyze these data by linking transcription factor expression levels to accessibility at individual regulatory sites. We have acknowledged the simplicity of the analysis in the text:

“The simple regression framework provided in this work is only a small step toward more complicated models that capture other relevant sources of heterogeneity.”

We have added text clarifying the presence of TTG2 binding sites from DAP-seq data for the peaks identified in this analysis:

“Using DAP-seq data for *TTG2*, we examined whether peaks with either positive or negative associations contain *TTG2* binding sites.²⁷”

This text accompanies the data presented in Figure 4D, where every peak was examined for the presence of a DAP-seq-verified *TTG2* binding site.

7) The manuscript overall looks like it was rather hastily put together, which made it unnecessarily confusing in a lot of places. The supplementary figures are cited wildly out of order. There are places where acronyms or abbreviations aren't clarified (e.g., c/e pre should be defined in Fig. 1 legend). Figure titles on the figure pages in a few places don't match the titles on the figure legend pages.

We apologize for these errors and thank the reviewer for their careful reading of the manuscript. We have reordered the Supplementary Figures, clarified abbreviations in the legends, ensured that figure titles are consistent throughout, and made language more consistent throughout with the goal of reducing confusion.

There are no Methods to describe the endoreduplication analyses.

Leaving out a more detailed description of this analysis was an error on our part; the Methods section has been updated with a detailed description of the endoreduplication analysis. See above comment #3 for new Methods text.

Line 505: software version missing. Data were not made available to reviewers for independent analysis.

The software version has been added, and an R object containing the scATAC and predicted expression data for each cell is available at this Drive link: <https://drive.google.com/file/d/1Xl1a1LMKUOWMwJRbpGIP-vh7F4iA9zdf/view?usp=sharing>

We will ultimately use Dryad, our lab website, or another data repository to host this data.

Most of these individually are minor issues, but collectively they make reading and reviewing the paper extremely challenging. Please take some time to check the overall organization of the paper.

Minor

1) Section starting Line 98: what percentage of peaks from snATAC are proximal

versus distal? Was either class (proximal or distal) enriched among those that are cell type specific?

We appreciate this suggestion: we found that 2,159/22,749 of the peaks resided more than 400 bp from their nearest gene, representing ~10% of the total peaks identified. Of these, 707 were found among the cell type-specific peak list (Supplemental Table 1). Although this is a minority of the total unique marker peaks ($707/4389 = 16.1\%$), this number is greater than what would be expected based on the fraction of distal peaks in the data set as a whole, indeed suggesting enrichment (16% of specific peaks vs 9.4% of total peaks). We have added text to the manuscript to highlight this satisfying finding:

“Though only 16% (707/4,389) of cell type-specific peaks were found to be distal, or greater than 400 base pair from the nearest gene, this was greater than the fraction expected by chance. Only 9.4% (2,159/22,749) of all peaks were distal, suggesting that these distal peaks are slightly (1.7x) enriched for regulatory sites that define cell identity.”

2) Line 134: is the inability to elucidate the identities of 3 clusters a snATAC-seq sensitivity issue? Do these have overall lower peaks numbers/UMIs than other clusters? Would sequencing the library deeper rescue these from obscurity and/or help deconvolve hair and non-hair epidermis?

We do not think that this is an issue of sequencing depth or UMI recovery. See boxplots below for the median UMIs by cluster from Figure S1E. In one case, one of these clusters shows *greater* UMIs than the median across clusters, suggesting that these cells may be doublets, another reason we avoided annotating in this case. The primary reason we felt that these clusters could not be annotated was the fact that they showed no marker genes when we examined differential gene activity scores; Supplemental Table 3 has been added to show the results of this marker gene analysis.

3) Line 231-232: should this be cluster endo #1, rather than #3?

This was a labeling error; thank you for pointing this out. We have revised this text:

“We further used this co-embedded space to refine our earlier manual cell type annotations by transferring labels of neighboring scRNA cells onto the scATAC cells (**Figure S3A, S3B**; while most of these labels matched, the greatest number of mismatches was seen in endodermis sub-type 1.”

4) Line 257: is there a paper citation supporting BLUEJAY’s role in endodermis differentiation?

Thank you for requesting a citation here. An important citation for Moreno-Risueno et al. 2015, which shows BLUEJAY’s role in endodermis differentiation, was lost in the editing process. This citation has been added back into the manuscript:

“Endodermis subtype 3 showed strong accessibility at the *BLUEJAY* (AT1G14580) locus encoding a C2H2 transcription factor implicated in endodermis differentiation (**Figure 3B**),³³ as well as *MYB122* (AT1G74080) and other genes for phenylpropanoid biosynthesis (*PER22*, *PER32*, *PER72*, *BGLU32*).³²”

5) One of the reasons this paper is challenging to follow and read is that the t-SNE/UMAP embeddings can change abruptly throughout the text, making it impossible to compare related information between different figures. Or different UMAPs are being used to support statements that they probably shouldn’t be used to

support. For example, Figure 3 shows one version of the snATAC-seq embedding with all cells or just zoomed to endodermis. Figure S3C shows a different embedding combining snATAC and scRNA-seq for endodermis cells. For the right half of Figure S3C (the gene expression plots), this would be clearer to understand in relation to the 3 endo clusters if nearest neighbor expression was shown in the version of the UMAP/t-SNE shown in Figure 3. Figure S4B shows yet another UMAP version to justify the lack of difference between cell cycle state between the 3 endo clusters, but the 3 endo clusters aren't even distinguishable on this version of the plot because they aren't seen in the scRNA-seq data. How can this actually support this statement?

We added shading to Figure 3 to clarify that the subsequent panels are a zoomed-in subset of these cells.

In Figure 3C, we should have been clearer that this is not a combined embedding, but scRNA-seq cells alone. The goal of this figure it to make the point that these genes would not have marked separate clusters or sub-groups of endodermis types using scRNA-seq data alone. We have revised a label in the panel and the accompanying text to make this clearer:

“The small set of marker genes identified for each scATAC subtype showed no consistent expression pattern in the scRNA-seq data (**Figure S5A**), suggesting that other features distinguished these three subtypes.”

The UMAP in Figure S4B (now S6B) is comprised of cells from all previously published root scRNA-seq experiments. Thank you for bringing this to our attention, as the goal of this panel was not clear in the text. This visualization exists primarily to show that proliferating cells do not mark a certain cluster or sub-group of the endodermis cells. It is an admittedly crude way to assess this, which is why we used the transferred gene expression procedure (explained in Figure S3C) to address the level of cell-cycle signature expression specifically in the 3 endodermis clusters. We have revised the text to make this clearer:

“This signature score identified cycling cells in other cell types, such as early epidermis cells near the quiescent center (**Figure S6A, S6B**) in a meta-analysis of previously published scRNA-seq data. However, when this signature score was transferred to the scATAC-seq endodermis clusters by the nearest neighbor procedure described in Figure S3C, we observed no differences corresponding to each endodermis subtype (**Figure S6C**).”

6) Line 263: what is it then that distinguishes the clusters? Are these peaks possibly marking some other noncoding sequence function other than positive (promoter/enhancer) regulation?

It is quite possible that the scATAC-specific peak regions contribute to something other than increased expression, as these genes do not show strong specificity within endodermal cells in scRNA-seq data (Figure S3C). We have revised the text to reflect this possibility (see above comment).

7) Paragraph Line 276: is there a series of typos in this paragraph, or is it really a paragraph about epidermis in a section otherwise devoted to endodermis? If it is about endodermis, the result is completely expected, isn't it, given that everything except the precursor cells should no longer be dividing?

Thank you for making us aware of this typo; this paragraph has been revised to refer to the endodermis. We agree that a difference in cell division may not be expected after the earliest stages of endodermis development, but we considered cell cycle a pertinent factor to rule out, as "new" clusters or cell states often appear in single cell RNA-seq data due to differences in proliferation.

8) Line 295-296: the plot shown in Figure 3C is completely different than Figure S3F. How exactly are these trends the same?

The confusion in Figure S3F (now S5F) is our fault. Unlike the RNA-based metric, the ATAC-based metric is represented there as raw complexity, meaning higher values are expected for less progressed endodermis cell types. To make this visualization more coherent, we have revised the y-axis of Figure S5F to show (1 – ATAC-based complexity), so the value increases with developmental progression, as in Figure 3C.

9) Paragraph p303: The information in this paragraph pertaining to non-endodermis cell types would be better suited for the earlier discussion of endoreduplication.

We agree with the reviewer's assessment here, and have decided to increase clarity by restructuring and moving this text to the earlier section discussing endoreduplication. The updated text is provided below:

"We then used a second method to identify endoreduplicated cells with a transcriptional signature. Instead of relying on the number of transpositions in the accessibility data directly, we instead leveraged the dataset integration described above (**Figure S3C**) to transfer scRNA-seq-based annotations to the cells in our scATAC experiment. To identify endoreduplicated cells in scRNA-seq data, we used a published set of marker genes for ploidy to generate signature scores for 2n, 4n, 8n and 16n ploidies.¹⁹ With these scores, we predicted endoreduplicated cells by calculating, for each cell, the ratio of the 8n signature relative to the diploid signature. Similar to the accessibility-based metric, this transcription-based approach identified endoreduplicated root cells in the expected pattern, with higher fractions in the

epidermis cell layer and diminished levels in the stele (**Figure S4D, S4E**). We found these two methods of identifying endoreduplicated cells to be concordant (**Figure S4F**), but because the accessibility-based classification was less quantitative, we used the transcriptionally-based metric in subsequent analyses. This metric captured an abundance of tetraploid xylem cells in the stele (**Figure S4E**), consistent with previous findings.¹⁹

Reviewer #2 (Remarks to the Author):

The authors present an interesting study combining new scATAC-seq data and published scRNA-seq data on the *Arabidopsis thaliana* root to investigate whether scATAC-seq can be used to differentiate cell identities, and whether the combination of single-cell RNA-seq and ATAC-seq data enhances the resolution of the cell regulatory landscape in roots. To my knowledge, this is the first scATAC-seq study in plants. Interesting results include the observation that chromatin patterns are more cell type-specific than previously thought, that scATAC-seq can distinguish cell types and in some instances developmental stages of cell types, and that combination of scATAC-seq and scRNA-seq data can resolve regulatory processes influencing cell type specification (with a case study on *TTG2*). Overall, I think this is an interesting and novel study, but I do have a few major concerns to be addressed.

Major comments

- The methods section of the manuscript needs to be amended substantially. For many of the computational analyses, there are no methods entries and the description in the results section is not sufficient to understand what was done exactly. For instance, the authors use a cutoff on the mean number of ATAC cuts per peak across cells to classify cells as endoreduplicated or not, but it is unclear what criterion this cutoff was based on.

We have added text to the Methods and main text to better clarify the scATAC-based endoreduplication metric (see above Reviewer 1, comment #3) and other Methods as described below.

Similarly, it is unclear how the $8n/2n$ and $4n/2n$ scRNA-seq measures for endoreduplication were calculated, which genes were involved in these calculations, or why these ratios were chosen and not the $16n/2n$ ratio or the $16n/4n$ ratio. Which ratios give information about which endoreduplication levels ?

We selected the $8n$ ratio because we thought that it should capture differences in the early to intermediate cell states of the endodermis, which were a focus of this paper. That said, we have also provided the $16n/2n$ ratio, and $16n/4n$ ratios as a resource for further investigating higher endoreduplication levels in the data.

Other methods statements in the results that do not give enough information on what was done include line 271-274 'we added annotations for cell cycle, developmental progression and endoreduplication to the combined root scRNA-seq data and used data

integration (as in Figure 2) ...', line 276-277 'we used a list of known cell-cycle marker genes to generate a signature score marking proliferating cells (Arabidopsis.org).', line 283-285 'We assessed developmental progression with two orthogonal methods: (i) correlation with published bulk expression data taken along longitudinal sections of the root and (ii) a modified measure of loss in transcriptional diversity (see Methods)' (for which there is no methods entry), and others.

The approach used to generate the transcriptional signature scores for endoreduplication were similar, and we have added text to describe our approach in greater detail in the Methods:

"Transcriptional diversity metric for developmental progression

Using the general premise that the number of unique genes expressed (transcriptional complexity) tends to be reduced across the developmental trajectory of a cell type as it moves from earlier to later stages,¹³ we devised a metric to approximate relative differences in developmental progression among cells. Measuring the number of unique genes expressed is distinct from measuring the number of UMIs or transcripts captured per cell, which can vary across cell types. To account for differential recovery of UMIs across cells in the transcriptional complexity measure, we modeled as a Loess fit the relationship between total UMIs captured and the number of unique genes expressed per cell. With this fit, we identified cells that have many more or fewer unique genes expressed than would be expected for cells over a range of captured UMIs. Developmental progression for each cell was defined as the residual of each point in this fit, allowing separation of earlier cells (more unique genes expressed than would be expected for a given number of captured UMIs) from later cells (fewer unique genes expressed than would be expected for a given number of captured UMIs)."

On line 156-157, the authors mention that they used a set of representative motifs for all *A. thaliana* TF gene families and nearly every individual transcription factor, but I don't see any analyses involving motifs for individual TFs. It is not clear how many and which motifs exactly were included in the analysis.

We have added text to the results explaining why individual TFs were aggregated on the family level to display final results, and provided additional text in the Methods to explain the source of the motif counts from individual TFs:

Results

“However, because motif sequence content for individual transcription factors is redundant, we computed means across each transcription factor family.”

Methods

“Motif analysis

Position weight matrices from the comprehensive DAP-seq dataset²⁷ were used as input into FIMO⁴⁶ to search for significant matches for each individual TF motif (adjusted p-value threshold < 1e-5) in each of the scATAC peaks. With the output of this motif scan, we generated a matrix that tallied counts of each individual motif within each peak. Each individual motif in the DAP-seq dataset²⁷ has an associated TF family, and the counts per peak were averaged by family.”

- Interpretation of the results is complicated by the fact that the scATAC-seq data and scRNA-seq data were not generated on the same cells in the same experiment (a study in mammalia where scATAC-seq and scRNA-seq are done simultaneously came out recently : <https://doi.org/10.1101/gr.257840.119>). An additional complication is that the authors use whole roots for scATAC-seq while the RNA-seq data they compare to (Ryu et al. 2019 <https://doi.org/10.1104/pp.18.01482>) was generated on root tips. In other words, most of the cells profiled by ATAC-seq should be older than those profiled in the scRNA-seq dataset. This may explain to some extent the lack of correspondence in e.g. marker genes between both datasets. Definitive answers on the correspondence between scATAC-seq and scRNA-seq data in plants will have to wait until both are generated on the same cells of the same samples.

We appreciate the insight of the reviewer in this comment. First of all, we agree that the approach presented in this work is premised on the assumption of cell states that can be correlated across experiments; we have added text to the Discussion to explain this:

“Technical differences in nuclei versus cell-based assays, size selection, developmental stage, and sequencing depth may also contribute to differences between scRNA and scATAC data sets. While increasing the depth of our ATAC signal per cell may alleviate some of this noise, we argue that the poor correlation between chromatin accessibility and gene expression is not a function of data quality.”

Ultimately, we can only account for these differences in a co-assay experiment. We expect that a co-assay experiment in Arabidopsis roots would, as the reviewer predicts, reveal more corresponding markers between these two data types, and have added text to this point:

“Although this single-cell ATAC study discovered many more dynamic accessible sites, the correlation between dynamic accessibility and gene expression in single cells

remained poor, reminiscent of the equally poor correlation seen in bulk studies. These data types would be integrated more faithfully in a true co-assay experiment.^{25,42}

Another possible reason for the observed relative lack of correspondence between chromatin accessibility and gene expression might be the inherently noisy nature of single-cell datasets, not only due to cellular stochasticity but also due to technical effects (low coverage per cell). The authors should comment on this in the discussion.

The median number of reads per cell in this study [median UMIS per cell = 7290] is comparable to previous scATAC studies in mammalian cells [Mezger et al 2018, median UMIs per cell = 8100], and is similar to recent large-scale scATAC atlases of development [Domcke et al. 2020, median UMIs per cell = 6042] (full citations at end of document).

However, there is no doubt that increased coverage per cell and increased cellular coverage per tissue would remove some level of noise from the relationships between expression and chromatin accessibility. We have highlighted this with new text in the Discussion:

“While increasing the depth of our ATAC signal per cell may alleviate some of this noise, we argue that the poor correlation between chromatin accessibility and gene expression is not a function of data quality.”

- on line 185-186, the authors state that ‘in contrast to scRNA-seq data, scATAC-seq data can provide insight into DNA copy number and its impact on gene regulation.’ On the other hand, in the following sections they establish that scRNA-seq derived predictions of DNA copy number (endoreduplication levels) are in fact superior. This begs the question whether there is added value to using scATAC-seq for predicting endoreduplication levels of cells.

The reviewer notes a disappointing truth that we have converged on also: scRNA-seq data are indeed superior for most basic needs of a single-cell genomics experiment. However, we feel the methodology for using scATAC data to predict copy number constitutes added value, even if a transcriptional metric currently outperforms the method. We hope our description of scATAC-based endoreduplication analysis conveys the method clearly enough that other researchers take notice. We expect future chromatin profiling technology may provide deep enough coverage to confidently predict copy number from scATAC data alone. We have added text to highlight this in the Discussion:

“Deeper coverage in future datasets should enhance our ability to detect rare cell types and more confidently predict copy number from accessibility data alone.”

- The manuscript convincingly shows that the scATAC-seq data captures three distinct endodermis developmental stages. Could the authors elaborate on why such subclusters would only be detectable for endodermis and not for other tissues? The authors further state on line 234-236 that ‘The three distinct ATAC clusters that were assigned an “endodermis” label with this approach are a striking example of scATAC data yielding greater stratification of cell types than the generally richer scRNA data.’ It is however worth noting that there are methods available to infer developmental trajectories within a cell type from scRNA-seq data, see e.g. <https://doi.org/10.1038/s41587-019-0071-9> .

We appreciate the utility of graph-based methods to infer developmental trajectories from scRNA-seq data. Our choice to use an orthogonal metric of developmental progression based on transcriptional complexity was intentional, in response to the many studies that have identified trajectories in single-cell genomics data that represent a continuum of cell states for a particular biological process rather than a developmental process. The premier example are cells undergoing the cell cycle which manifests as a trajectory in two-dimensional embeddings. In short, we think that the transcriptional complexity metric is more conservative and does not depend on the graph structure of the two-dimensional embedding, which might be driven by a biological process like the cell cycle.

We have text in the manuscript addressing this possibility:

“Structure within two-dimensional embeddings of scRNA-seq and scATAC-seq data derived from developing tissues is often associated with differences in developmental progression or other asynchronous processes like the cell cycle.”

Minor comments

- the ordering of manuscript sections and figures is sometimes confusing. The section on endoreduplication starting at line 183 for instance refers to ‘orthogonal measurements of endoreplication’, referring to scRNA-seq-based inferences that are only discussed several pages further on.

We agree with this comment, which matches a previous reviewer comment. The text of the manuscript has been restructured to correct this confusion. The new text is found under the early section “**Epidermal cell layers show increased levels of endoreduplication.**”

- on line 212-213, the authors state that the root tissue sampled for the scATAC-seq experiment and previous scRNA-seq experiments was similar, while the methods state that whole roots of 12 day-old seedlings were used, as opposed to root tips of 5 day-old seedlings in Ryu et al. 2019 (<https://doi.org/10.1104/pp.18.01482>).

The root tissue in the two sets of experiments was broadly similar, but we have revised the text to highlight that the similarity of the datasets is qualitative, rather than quantitative. Some more mature cell states in the major root cell types may be underrepresented in the Ryu et al 2019 data, which were derived from younger root tips.

“Although the root tissue sampled for our scATAC-seq experiment was not identical to that used in previous scRNA-seq experiments, we expected that the same major cell types were sampled in both experiments.”

- Some of the references, in particular in the methods section and figure legends, are not in the bibliography and are formatted differently, e.g. Jean-Baptiste et al. 2019. Some other references are missing, e.g. for the gene functions discussed in line 248-258.

We have corrected the references mentioned here.

We have also added references for the gene functions.

Suberization function of FAR1, FAR4, and FAR5:

Domergue, F., Vishwanath, S. J., Joubès, J., Ono, J., Lee, J. A., Bourdon, M., ... & Rowland, O. (2010). Three Arabidopsis fatty acyl-coenzyme A reductases, FAR1, FAR4, and FAR5, generate primary fatty alcohols associated with suberin deposition. *Plant Physiology*, 153(4), 1539-1554.

BLUEJAY:

Moreno-Risueno, M. A., Sozzani, R., Yardımcı, G. G., Petricka, J. J., Vernoux, T., Blilou, I., ... & Benfey, P. N. (2015). Transcriptional control of tissue formation throughout root development. *Science*, 350(6259), 426-430.

Functional identities for other genes were determined using gProfiler, for which we have also added a reference:

Raudvere, U., Kolberg, L., Kuzmin, I., Arak, T., Adler, P., Peterson, H., & Vilo, J. (2019). g: Profiler: a web server for functional enrichment analysis and conversions of gene lists (2019 update). *Nucleic acids research*, 47(W1), W191-W198.

- Supplementary figure 3 : panel D is not discussed in the legend. Also ‘subtype 3’ is mentioned underneath panel C but it is unclear why. In panels E and F, it is unclear whether the differences observed between the bar chart means are statistically significant. In panels H, I and J, color legends are missing.

We appreciate the reviewer’s attention to these errors, which have now been corrected.

Figure S3 is now Figure S5.

Figure S3D (now S5E):

“Boxplots showing data from Figure 3C, with average developmental progression computed with different numbers of nearest neighbors. Above each plot, the number of neighboring cells from the scRNA-seq data used to predict developmental progression of each scATAC endodermal cell is shown. The relative differences in predicted developmental progression is insensitive to the number of nearest neighbors used in the procedure.”

Figure S3C (now S5A) has been updated to explain the ‘subtype 3’ label:

“Expression of genes whose accessibility marked endodermis subtype 3 are shown in endodermis cells from published scRNA-seq experiments. Despite the specific accessibility patterns of these genes in endodermis subtype 3, expression patterns in cells derived from scRNA-seq are variable.”

In Figure S3F and S3F (now S5B, and S5H), stars indicating statistical significance have been added, and explained in the accompanying legend.

In Figure S3H, S3I, and S3J (now S5C, S5D, S5I) color legends have been added.

Reviewer #3 (Remarks to the Author):

In this manuscript, Dorrity et al. report the measurement of single cell chromatin accessibility in Arabidopsis roots using ATAC-seq. Chromatin accessibility is the gatekeeper of transcription factor binding. This study is important since, as outlined by the authors, recent studies uncovered overall invariable chromatin accessibility in different plant organs, which could reflect cell type heterogeneity or important regulatory layer of plant transcription plasticity.

Analyzing 5283 single nuclei resolved nine distinct cell clusters. The identity of these cell populations was assigned combining TSS-proximal accessibility, reflecting gene activity, with scRNA-seq profiles. Similarly to the whole organ level, TSS-proximal accessibility did not show high correlation with gene expression profiles. Nevertheless, combining scATAC-seq with scRNA-seq allowed focusing on transcription factors and their combinations that are unique to different cell types.

The authors identified three clusters of endodermal cells in the scATAC-seq data. The loss in transcriptional diversity in these clusters led to suggest that they were related to different developmental stages. The progression of developmental states in these clusters and also in epidermal cells was further supported by elevated endoreduplication.

This study is important for pioneering measurements of sc-chromatin accessibility in plants. On the scientific merit, the study illuminates the level of chromatin accessibility specificity in different cell types and apply high-end tools to analyze the data and uncover novel potential transcription factors that regulate the genetic programs in these cells.

The data quality seems to be high and well presented.

Minor points to be addressed:

1. The lack of correlation between chromatin accessibility at gene proximal loci and gene expression is expected. However higher correlation was recently shown between gene expression and distant organ-specific accessible loci (ATAC-seq studies from Deal and Hakim labs). Please test this possibility in your data.

We come to a similar conclusion as the aforementioned studies: despite the overall poor correlation between expression and chromatin accessibility, the few loci that show concordance in specific organs or cell types are sufficient to align these distinct data types. We show this primarily through our co-embedding procedure (Figure 2), wherein cell type labels transferred from aligned scRNA-seq data matched well with manual annotations of scATAC-seq data alone (Figure S3).

Furthermore, we have added an analysis showing that root hair and root non-hair specific ATAC-seq data generated with the INTACT method shows concordance with pseudobulked scATAC-seq data generated by averaging accessibility across all cells with an “epidermis” label. The results of the analysis are now in Figures S1C and S1D, and are reproduced below. Given that INTACT-based approach produces results similar to the scATAC-based approach, we expect that the previously identified organ-specific accessible loci are partly responsible for the effectiveness of the alignment procedure mentioned above.

2. scRNA-seq data (references 2,5,6 in the manuscript) uncovered 9 to 15 distinct cell clusters and nine distinct cell clusters were resolved in this study. Please consider revising the statement in lines 235, 236 accordingly. In addition, the cell identities of the clusters from both technologies were not identical (i.e. the different endodermis clusters were not uncovered in scRNA-seq). Please discuss the possible reasons for the differences between the two technologies, in addition to the number of analyzed cells. It would be useful if the authors could provide a quantitative view of the possible effect of the number of analyzed cells on the expected number of cell types to be uncovered by scATAC-seq.

We note the reviewers concern with the overall lower number of distinct cell types in scATAC vs. previous scRNA data and have added a sentence to clarify the statement in lines 235, 236, which was focused on the surprising example of scATAC capturing more “types” within the endodermis.

See revised sentence below:

“The three distinct scATAC clusters that were assigned an “endodermis” label with this approach are a striking example of scATAC data yielding, within a single cell type, greater stratification of “types” than the generally richer scRNA data.”

We have characterized possible sampling biases in our previous scRNA-seq study of the Arabidopsis root (see Figure 1F of Jean-Baptiste et al. 2019), which contained fewer cells (3,121 cells) than the present study (5,283 cells). Accordingly, we expect a similar overall sampling of cell types in the scATAC data as we did with scRNA data.

3. A graphical map of the different root cell types (such as Fig. 1G in PMID: 30718350) would help non-plant scientists navigating the study.

We have added a simple graphical map to Figure S2A, reproduced below:

stele
pericycle
endodermis
cortex
epidermis

4. Please name the supplemental tables.

We have added names to each of the Supplemental Tables:

Supplementary Table 1 – MarkersByCluster_peaks (ATAC-peak markers of cell type)

Supplementary Table 2 – MarkersByCluster_geneActivity (gene + promoter markers of cell type)

Supplementary Table 3 – ATAC_markerOverlap (cell-type specific ATAC genes matching cell type specific RNA genes)

Supplementary Table 4 – endodermis1_predictedExpression_markers (gene expression markers inferred from scRNA-seq neighbors for Endodermis sub-type 1)

Supplementary Table 5 – endodermis2_predictedExpression_markers (gene expression markers inferred from scRNA-seq neighbors for Endodermis sub-type 2)

Supplementary Table 6 – endodermis3_predictedExpression_markers (gene expression markers inferred from scRNA-seq neighbors for Endodermis sub-type 3)

Supplementary Table 7 - epidermis_cortex_pre_TTG2_fitmodelsOutput (ATAC-peaks whose accessibility shows significant [$q < 0.05$] association with inferred TTG2 gene expression)

Reviewer #4 (Remarks to the Author):

This is a first study to employ single cell ATAC seq to understand the gene regulatory programs in the Arabidopsis root, an important model system for plant development.

The study is overall very straightforward in applying existing experimental and

computational tools. There are a couple of concerns that may impact the observations and conclusions:

- Perhaps there are so few cell type-specific regions because the authors are summarizing ATAC signal in gene bodies, and in turn include only up to 400nt upstream. Previous reporter gene studies typically take much larger upstream regions to recapitulate transcription (3kb).

So, on the one side, this will likely introduce noise since the majority of gene body space does not represent regulatory elements with dynamic accessibility, while excluding most of the genome that is thought to harbor such elements. Why not summarise at the gene level by assigning each peak to its closest (downstream) gene? To support the gene body quantification, in addition to the regulatory region browser shots in fig1, the authors should make a similar panel with representative dynamic gene bodies. Also, the authors should show pseudo-bulk browser track comparisons to published tissue isolations from reference 11 referred to on line 121.

We have generated representative example coverage plots of gene body accessibility differences, shown below (provided for clarity, does not appear in final manuscript):

In lieu of the more qualitative comparisons to published tissue isolations by browser tracks, we have conducted a more thorough analysis of accessibility patterns from

these studies using peaks identified in the present scRNA-seq study. These results (also discussed in Comment 1 for Reviewer #1 and Reviewer #3) show that epidermis accessibility from scRNA-seq is higher in peaks that show higher accessibility in INTACT-preps of epidermis cell types. As a control test, we found that this trend does not hold for accessibility in stele cells identified in our scATAC study. These figures are shown in Comment 1 above, and have been added as Figures S1C and 1D.

- the authors state on lines 295 and 296 that the atac-seq-based developmental progression score shows the same trends as the rna-seq-based developmental progression metric which, if true, would be an interesting observation. But examination of Supp FigS3F and S3G shows, if anything, a relative inverse trend (progressively lower scores for cell types 1, 2, 3 as opposed to progressively increasing for the rna metric), and an overall poor correlation. Perhaps this is due to a difference in the way the scores are derived, but there is not sufficient information in the methods to determine how each was calculated. Furthermore, on lines 311 and 312 the authors refer to S3F and S3G again, but what seems to be in the context of endoreduplication, which leaves the reader confused as to what is actually being plotted and compared to what.

We thank the reviewer for pointing out the lack of clarity in Fig S3F. Unlike the RNA-based metric, the ATAC-based metric is represented there as raw complexity, meaning higher values are expected for less progressed endodermis cell types. To make this visualization more coherent, we have updated the y-axis of Fig S3F (now Fig S5F) to show $(1 - \text{ATAC-based complexity})$, so the value increases with developmental progression.

We have also added additional text to the Methods to more completely describe both the endoreduplication and developmental progression metrics. We hope this clarifies the relationship that the reviewer has highlighted here.

Overall, the presentation of the developmental progression metrics in the text needs to be clearer and the description in the methods of how each was calculated is insufficient.

A full description of the developmental progression metric was indeed lacking, and a more comprehensive description has been added to the Methods (see Reviewer 2, Major Comment 1).

The statement about the utility of ATAC quantification for determining developmental progression likely needs to be revised unless otherwise clarified.

- the number of cells and reads per cell is quite on the low side for the current state of

the art. A calculation how far from saturation this coverage is would provide a relevant point for discussion.

Shown below is a quality control plot to show the level of saturation in our libraries, which were sequenced to a 21% duplication rate (panel is not featured in final manuscript). The median number of reads per cell in this study [7290] is comparable to previous scATAC studies in mammalian cells [Mezger et al 2018, median UMIs per cell = 8100], and is similar to recent large-scale scATAC atlases of development [Domcke et al 2020, median UMIs per cell = 6042] (full citations at end of document).

We have also added a note about the cellular coverage and saturation to the Discussion, addressing how this might be improved for future studies:

“In the future, single-cell studies of more complex plant tissues in crops and other species will necessitate larger numbers of profiled cells and higher numbers of cuts per cell. Deeper coverage in future datasets should enhance our ability to detect rare cell types and more confidently predict copy number from accessibility data alone.”

While the number of captured nuclei in this study is comparable to previous²⁻⁶ and current scRNA-seq studies [Wendrich et al. 2020, *Science*], we are likely under sampling the true number of distinct chromatin states associated with specific cell types. For example, a recent large scRNA-seq study combined several datasets to profile 110,000 cells [Shahan et al 2020, *bioRxiv*] identified more vascular cell types than the 3 stele-associated cell types identified in this study. At this point, we cannot distinguish whether a difference like this is due to total numbers of cells, or differences in the way information on cell type identity is encoded in scRNA and scATAC data.”

References

Mezger, A., Klemm, S., Mann, I., Brower, K., Mir, A., Bostick, M., Farmer, A., Fordyce, P., Linnarsson, S. and Greenleaf, W., 2018. High-throughput chromatin accessibility profiling at single-cell resolution. *Nature communications*, 9(1), pp.1-6.

Domcke, S., Hill, A.J., Daza, R.M., Cao, J., O’Day, D.R., Pliner, H.A., Aldinger, K.A., Pokholok, D., Zhang, F., Milbank, J.H. and Zager, M.A., 2020. A human cell atlas of fetal chromatin accessibility. *Science*, 370(6518).

Wendrich, J.R., Yang, B., Vandamme, N., Verstaen, K., Smet, W., Van de Velde, C., Minne, M., Wybouw, B., Mor, E., Arents, H.E. and Nolf, J., 2020. Vascular transcription factors guide plant epidermal responses to limiting phosphate conditions. *Science*, 370(6518).

Shahan, R., Hsu, C.W., Nolan, T.M., Cole, B.J., Taylor, I.W., Vlot, A.H.C., Benfey, P.N. and Ohler, U., 2020. A single cell Arabidopsis root atlas reveals developmental trajectories in wild type and cell identity mutants. *bioRxiv*.

REVIEWER COMMENTS

Reviewer #1 (Remarks to the Author):

Overall, the manuscript is much improved over the initial submission, and most of my comments have been suitably addressed. However, I still have a major concern about the analysis and interpretation of the endodermis developmental trajectory.

Major

Figure 3 and related text: While I agree with the authors' expectation that technical reasons are likely to limit the number of very mature endodermis cells captured in their analysis, it doesn't fully explain why their endodermis development analysis appears to be exactly reversed relative to 1) what is known about endodermis development and 2) their own cell cluster analysis in Figure 1. If I understand correctly, the authors conclude that endo 1 is the least differentiated, then endo 2, and endo 3 is most differentiated. However, this remains confusing for a number of reasons:

1. The data presented in Figure 1A would suggest that endo 3 is the least differentiated, as it appears next to the c/e precursor cluster in the UMAP.
2. The data presented in Figure 1D would suggest that Endo 3 is the least differentiated, as the c/e pre transcription factor motif signatures look more closely related to endo 3 than either endo 1 or 2.
3. The literature on the marker genes mentioned for each cluster in the section "scATAC-seq captures three distinct endodermis types representing different developmental stages" would suggest that the developmental ordering is reversed. Endo 1 is described as having specific accessibility of genes involved in suberization and Casparian strip formation. Endo 2 also has high accessibility of Casparian strip genes. Endo 3 is marked by MYB36 accessibility. Given what is described in the literature about endodermis development, these genes would strongly suggest that endo 3 is early, endo 2 is middle, and endo 1 is late endodermis. MYB36 is known to be an important regulator of early endodermis differentiation (PMID: 26371322), and a recent very high-resolution scRNA-seq of Arabidopsis root shows that it is expressed prior to the CASP genes along the endodermis lineage (biorxiv doi: <https://doi.org/10.1101/2020.06.29.178863>). In endodermis development, Casparian strip formation occurs in maturing endodermis and is then followed by suberization, which is something of the "hallmark" of fully mature endodermis (PMID: 26343015).

How do the authors account for their endodermis developmental progression analyses in light of these apparent contradictions?

Minor

1. Line 74: Claim of novelty ("first single-cell resolution...") should be removed or toned down in light of similar, published data set (PMID: 33422696).

Reviewer #2 (Remarks to the Author):

Most of my comments were addressed satisfactorily. I do have a few remaining comments related to methodological uncertainties on the scRNA-seq-based endoreduplication analysis :

- For the method to identify endoreduplicated cells based on scRNA-seq data, I did not find any marker gene sets for specific ploidy levels in the Bhosale et al. (2018) Plant Cell paper. I assume the marker gene sets used were subsets of the 332 marker genes used in this paper that peak in expression at specific endoploidy levels ? This should be explicitly mentioned.
- Were the marker gene expression levels normalized per gene in some way (e.g. z-scored) before

calculating the average expression of marker groups (which I assume are the signature scores) and the 8n/2n endoreduplication scores? If not, this should definitely be done to avoid bias caused by e.g. some 8n marker genes having much higher expression levels than 2n marker genes even in diploid cells.

- the use of specific ratios to call specific ploidy levels is tricky, e.g. in Supplementary Figure 4 where a high 4n/2n ratio is taken to be indicative of a 4n ploidy level. It is possible that some cells with a high 4n/2n ratio might have an even higher 8n/2n ratio and are therefore likely 8n rather than 4n. Similarly, the 8n/2n ratio might miss endoreduplicated cells if the 4n/2n ratio of the cells is high but the 8n/2n ratio is not much different from the 2n ratio ($2n/2n = 1$).

- a last point related to these scores: the color scales for all ratio scores in the figures seem to be centered around 0. Are these log scores?

- with regard to a previous comment on the sentence 'In contrast to scRNA-seq data, scATAC-seq data can provide insight into DNA copy number and its impact on gene regulation': this sentence is still there, and it is still at odds with the fact that the authors also use scRNA-seq data in addition to scATAC-seq data to infer cell endoreduplication levels. Please reformulate.

Reviewer #3 (Remarks to the Author):

My comments have been addressed by the authors. The manuscript clarity has much improved.

Reviewer #5 (Remarks to the Author):

In their revised manuscript, the authors have partially satisfied my concerns. With regards to capturing regulatory accessibility dynamics with gene activity scores, the authors have included an analysis of distal peaks showing cell-type specific differential accessibility. While this is a useful analysis, it does not consider that regulatory elements themselves will only make up a fraction of ATAC signal in gene bodies and the rest is likely noise that could potentially overwhelm any useful signal. While the authors correctly discuss the many technical and biological issues that could lead to discrepancies between scATAC and scRNA-based cell clustering and integration, they seem to glance over a major unsolved problem in the field which is appropriate summarising of regulatory element accessibility dynamics at the gene level. Since many groups are working on this exact problem, the authors should at least list it in the discussion as a potential complication.

With regards to developmental progression ordering based on ATAC data, the authors have included a better description of the RNA-based complexity method used, but there is no mention in the text or methods about how this is calculated using the ATAC signal. I found the following statement regarding this in the legend for Supp. FigS5 "analogous to transcriptional complexity metric from Fig. 3C, only computed as total number of accessible genes rather than total number of transcribed genes", but a more detailed description must be included in the methods section.

REVIEWER COMMENTS

Reviewer #1 (Remarks to the Author):

Overall, the manuscript is much improved over the initial submission, and most of my comments have been suitably addressed. However, I still have a major concern about the analysis and interpretation of the endodermis developmental trajectory.

Major

Figure 3 and related text: While I agree with the authors' expectation that technical reasons are likely to limit the number of very mature endodermis cells captured in their analysis, it doesn't fully explain why their endodermis development analysis appears to be exactly reversed relative to 1) what is known about endodermis development and 2) their own cell cluster analysis in Figure 1. If I understand correctly, the authors conclude that endo 1 is the least differentiated, then endo 2, and endo 3 is most differentiated. However, this remains confusing for a number of reasons:

1. The data presented in Figure 1A would suggest that endo 3 is the least differentiated, as it appears next to the c/e precursor cluster in the UMAP.
2. The data presented in Figure 1D would suggest that Endo 3 is the least differentiated, as the c/e pre transcription factor motif signatures look more closely related to endo 3 than either endo 1 or 2.

We understand the reviewer's intuition to equate proximity in the UMAP with proximity in cell identity and developmental stage. However, the UMAP algorithm generates a simplified, two-dimensional representation of a complex, high-dimensional space. While the cells near each other in the UMAP share similar transcriptomes, this relationship does not hold consistently at larger distances. Two clusters of cells near each other are not necessarily closely related, much like "endo 1" and "stele 2". While the distance between these clusters is similar in the UMAP, we would not conclude that there is a developmental relationship between endodermis and stele without some further evidence. Most analyses focusing on developmental progression within a particular cell type exclude cells with different developmental origin (known by some orthogonal evidence). These references provide good examples.²⁻⁶

Because proximity in UMAP space does not suffice to enable us to make conclusions about developmental stage, we conducted for this manuscript in-depth analyses of developmental progression¹³ and endoreduplication¹⁹. The results of these analyses suggest that "endo 3" is more advanced developmentally than "endo 1."

3. The literature on the marker genes mentioned for each cluster in the section "scATAC-seq captures three distinct endodermis types representing different developmental stages" would suggest that the developmental ordering is reversed. Endo 1 is described as having specific accessibility of genes involved in suberization

and Casparian strip formation. Endo 2 also has high accessibility of Casparian strip genes. Endo 3 is marked by MYB36 accessibility. Given what is described in the literature about endodermis development, these genes would strongly suggest that endo 3 is early, endo 2 is middle, and endo 1 is late endodermis. MYB36 is known to be an important regulator of early endodermis differentiation (PMID: 26371322), and a recent very high-resolution scRNA-seq of Arabidopsis root shows that it is expressed prior to the CASP genes along the endodermis lineage (biorxiv doi: <https://doi.org/10.1101/2020.06.29.178863>). In endodermis development, Casparian strip formation occurs in maturing endodermis and is then followed by suberization, which is something of the “hallmark” of fully mature endodermis (PMID: 26343015).

While accessibility is necessary for gene expression, it is by no means sufficient. In all eukaryotes examined to date, the correlation between accessibility and gene expression is surprisingly low. The genes described here are those with accessibility specific to each endodermis subtype. While we believe that this subtype-specific accessibility is biologically relevant, e.g. for priming subsequent gene expression or as a consequence of prior gene expression,²³ it cannot be fully equated with gene expression. Nevertheless, to address this further, we have provided an analysis of gene expression (predicted from ATAC-RNA dataset integration) in each endodermis subtype, reproduced from the main text here for convenience:

“The early endodermis type, which is not yet endoreduplicated, showed an enrichment of genes (**Supplementary Table 4**) involved in Casparian strip formation (*CASP3*, *CASP5*) and wax biosynthesis (HHT1). The intermediate sub-type 2 also showed enrichment for genes involved in Casparian strip formation (*CASP3*, *CASP4*, *CASP5*, *GSO1*), as well as mechanosensitive ion channels (*MSL4*, *MSL6*, *MSL10*) (**Supplementary Table 5**). The most advanced endodermis sub-type 3 showed enrichment for stress responses and metabolism of toxic compounds, kinase activity, and aquaporin water channels (**Supplementary Table 6**), consistent with this mature endodermis cell type modulating water permeability via aquaporins as well as through suberization.³⁴”

Given these gene expression patterns, the “endo 3” subtype is likely expressing the full complement of genes related to mature endodermis function, whereas the earlier “endo 1” subtype appears to express genes involved in the formation of the Casparian strip, a prerequisite for mature endodermis function, as noted by the reviewer.

There is no doubt that proper endodermis differentiation as described by the reviewer occurred in our sample, with MYB36 expression defining an early state. However, the three endodermis clusters we identified in our scATAC-seq data may not correspond exactly to this differentiation trajectory and its known gene expression patterns. To further explore this thought, we examined the MYB36 expression pattern, as inferred by neighboring scRNA-seq cells, in the UMAP – it is highest in the “endo 3” subtype, but not absent in “endo 1” or “c/e pre.” Thus, our results are less clear compared to the well-defined MYB36 expression in scRNA-seq clusters alone.

Furthermore, we took advantage of the invaluable description of endodermis differentiation from Shulze *et al.*, 2019 to examine additional genes that are expressed in early or late stages of endodermis development.³ Consider, for example, the late-expressed genes *SGN1* (AT1G61590) and *ESB1* (AT2G28670) shown below. These late-expressed genes appear to show the highest levels in “endo 3.” Evidence like this, combined with corroborating patterns of decreased transcriptional complexity and increased endoreduplication, suggest to us that “endo 3” represents, on average, a more differentiated endodermis subtype.

How do the authors account for their endodermis developmental progression analyses in light of these apparent contradictions?

The challenge to describe these scATAC clusters in terms of gene expression is at the very heart of an unresolved conflict in the gene regulation field. Accessibility and gene expression are correlated, which facilitates the annotation of our clusters to some extent. However, there is not a direct, one-to-one relationship between accessibility and expression at individual loci.⁸ Similarly, we do not expect there to be a one-to-one match for every scATAC cell cluster with one found in scRNA space.

We have added text to the Discussion to more explicitly highlight the seemingly “mixed up” endodermis differentiation trajectory, in addition to discussing the importance of interpreting accessibility data for what it is: a rough snapshot of complex gene regulatory processes that contribute to variation in gene expression observed across cell types. Because accessibility data have well-known limits, we have used all analytical tools at our disposal to describe the three endodermis clusters.

“We found three distinct endodermis cell clusters that differed in transcriptional complexity and endoreduplication, consistent with the clusters representing early, middle, and late stages of endodermis development previously characterized with scRNA-seq.³ However, at the level of individual marker genes, like the early-expressed *MYB36*, a more complex scenario emerged: accessibility of *MYB36* was similar across the three endodermis subtypes, and predicted expression of *MYB36* transcript, while present in all three subtypes, and, was highest in “endodermis 3”, the cluster with the most advanced developmental progression and highest endoreduplication. Without a direct, one-to-one relationship between accessibility and expression at individual loci, we do not expect a one-to-one match for every endodermis scATAC cell cluster with one found in scRNA space.”

Minor

1. Line 74: Claim of novelty (“first single-cell resolution...”) should be removed or toned down in light of similar, published data set (PMID: 33422696).

We have updated this sentence to tone down the claim of novelty and added the reference. We are very happy to have company in this endeavor; interpretation of scATAC-seq data in the root is considerably more complex than interpretation of scRNA-seq.

Reviewer #2 (Remarks to the Author):

Most of my comments were addressed satisfactorily. I do have a few remaining comments related to methodological uncertainties on the scRNA-seq-based endoreduplication analysis :

We appreciate the reviewer’s comments in making the endoreduplication analysis clearer for the reader.

- For the method to identify endoreduplicated cells based on scRNA-seq data, I did not find any marker gene sets for specific ploidy levels in the Bhosale et al. (2018) Plant Cell paper. I assume the marker gene sets used were subsets of the 332 marker genes used in this paper that peak in expression at specific endoploidy levels ? This should be explicitly mentioned.

Using the confidence metrics provided in the Bhosale et al. (2018) Plant Cell paper, we used the top 250 genes from each ploidy level to compute the signature scores. This is an important detail that was previously omitted. The main text has been updated to clarify this:

“To identify endoreduplicated cells in scRNA-seq data, we used a published set of marker genes, sub-setting the top 250 markers for each ploidy level to generate signature scores for 2n, 4n, 8n and 16n ploidies.¹⁹”

- Were the marker gene expression levels normalized per gene in some way (e.g. z-scored) before calculating the average expression of marker groups (which I assume are the signature scores) and the 8n/2n endoreduplication scores? If not, this should definitely be done to avoid bias caused by e.g. some 8n marker genes having much higher expression levels than 2n marker genes even in diploid cells.

This is another important clarification, as we have indeed computed signature scores based on normalized expression values. The Methods text has been updated to reflect this.

“In the first approach (as in **Figure 3D, Figure S4D, S4E, Figure S5B, S5I**), validated sets of endoreduplication markers for 2N, 4N, and 8N cells were used to identify endoreduplicated cells in the scRNA data.¹⁹ For these markers, we used the top 250 ranked genes ranked for each ploidy level. The signature scores were computed on normalized expression values for each gene using the `monocle3` function `aggregate_gene_expression`, producing a log-scale signature of ploidy.”

- the use of specific ratios to call specific ploidy levels is tricky, e.g. in Supplementary Figure 4 where a high 4n/2n ratio is taken to be indicative of a 4n ploidy level. It is possible that some cells with a high 4n/2n ratio might have an even higher 8n/2n ratio and are therefore likely 8n rather than 4n. Similarly, the 8n/2n ratio might miss endoreduplicated cells if the 4n/2n ratio of the cells is high but the 8n/2n ratio is not much different from the 2n ratio ($2n/2n = 1$).

We agree that the question of which ratio-based ploidy metric to use is unclear. In general, for the scope of this manuscript, we wanted a qualitative judgement of whether a cell's ploidy was above 2n, rather than a specific estimation of the degree of endoreduplication. For this purpose, we rely on the 8n/2n metric as a conservative estimate of whether a cell's ploidy is >2n. However, we highlight the example in the xylem (Supplemental Figure 4) because we have a specific expectation of that cell type's ploidy (4n).¹⁹ We believe that there is potential to more finely discern levels of endoreduplication, for example by including other features, such as the expected increase in DNA replication, but that these analyses deserve their own treatment in another manuscript.

- a last point related to these scores: the color scales for all ratio scores in the figures seem to be centered around 0. Are these log scores?

These are indeed log scores, and we have added this clarification to the Methods, along with the Monocle3 function used to produce the signatures (see above comment).

- with regard to a previous comment on the sentence 'In contrast to scRNA-seq data, scATAC-seq data can provide insight into DNA copy number and its impact on gene regulation' : this sentence is still there, and it is still at odds with the fact that the authors also use scRNA-seq data in addition to scATAC-seq data to infer cell endoreduplication levels. Please reformulate.

We agree that the highlighting here is awkward, as the scRNA-seq-based endoreduplication signature ultimately proves most useful later in the manuscript. Accordingly, we have toned down the suggestion that scATAC will clarify endoreduplication patterns:

"scATAC-seq data can potentially provide insight into DNA copy number and its impact on gene regulation, but this potential has not been thoroughly explored."

We offer this approach to examine endoreduplication with scATAC-seq data alone, since the scRNA-seq-based approach is only applicable in an organism for which we have a set of genes associated with particular ploidy.

Reviewer #3 (Remarks to the Author):

My comments have been addressed by the authors. The manuscript clarity has much improved.

We appreciate your effort in improving this manuscript.

Reviewer #5 (Remarks to the Author):

In their revised manuscript, the authors have partially satisfied my concerns. With regards to capturing regulatory accessibility dynamics with gene activity scores, the authors have included an analysis of distal peaks showing cell-type specific differential accessibility. While this is a useful analysis, it does not consider that regulatory elements themselves will only make up a fraction of ATAC signal in gene bodies and the rest is likely noise that could potentially overwhelm any useful signal. While the authors correctly discuss the many technical and biological issues that could lead to discrepancies between scATAC and scRNA-based cell clustering and integration, they seem to glance over a major unsolved problem in the field which is appropriate summarising of regulatory element accessibility dynamics at the gene level. Since many groups are working on this exact problem, the authors should at least list it in the discussion as a potential complication.

We realize that, in opting for the simplest summarization possible (all transpositions in a given gene and its proximal promoter region), we have missed an opportunity to discuss

the challenging nature of this problem. We now acknowledge this shortcoming and have added text to the Discussion:

“Furthermore, decisions made in relating transposition events to genes complicates analysis; by opting for a simplistic grouping of proximal promoter cuts with gene body cuts, rather than a more nuanced method,⁴² we may have introduced noise into the gene-level counts.”

We acknowledge that noise is a concern when analyzing the rare transposition events in single cells compared to bulk data. Transposition events outside the regulatory elements, or “peaks,” are included in our simple summarization at the gene level. However, we compared the quality of these data when pseudobulked to actual bulk data, and found them to be concordant, even in cases where we could analyze cell type-specific peaks (Supplementary Figure 1A-D). We found that the fraction of reads within peaks (or regulatory elements) was higher than our lab’s own bulk accessibility studies (Supplementary Figure 1F),²³ suggesting that the quality of the single-cell accessibility data minimizes some of the contribution of noisy transposition events to our gene activity scores.

With regards to developmental progression ordering based on ATAC data, the authors have included a better description of the RNA-based complexity method used, but there is no mention in the text or methods about how this is calculated using the ATAC signal. I found the following statement regarding this in the legend for Supp. FigS5 “analogous to transcriptional complexity metric from Fig. 3C, only computed as total number of accessible genes rather than total number of transcribed genes”, but a more detailed description must be included in the methods section.

We have added an explanation of the ATAC-based metric of gene complexity to the Methods:

“A similar analysis was conducted using the scATAC data alone, as previously described.¹³ By using the gene activity score matrix (summarizing all transposition events within each gene, and within 400bp of its start site), we computed for each cell the total UMIs normalized by the total number of accessible genes (count ≥ 1). Much like the transcriptional complexity metric above, we expect this ratio to increase with developmental progression. As the total number of accessible genes decreases with developmental time, a larger fraction of UMIs are found in this smaller set of genes.”

We want to reiterate that we were less impressed by the ATAC-based version of the developmental progression metric, but that its reduced dynamic range was consistent with observations in the original manuscript describing it compared to an RNA-based measure of complexity.¹³ We hope that this message comes through in the revised manuscript.

References

2. Jean-Baptiste, K. *et al.* Dynamics of gene expression in single root cells of *A. thaliana*. *Plant Cell* **31**, tpc.00785.2018 (2019).
3. Shulse, C. N. *et al.* High-Throughput Single-Cell Transcriptome Profiling of Plant

- Cell Types. *Cell Rep.* **27**, 2241-2247.e4 (2019).
4. Zhang, T. Q., Xu, Z. G., Shang, G. D. & Wang, J. W. A Single-Cell RNA Sequencing Profiles the Developmental Landscape of Arabidopsis Root. *Mol. Plant* **12**, 648–660 (2019).
 5. Ryu, K. H., Huang, L., Kang, H. M. & Schiefelbein, J. Single-cell RNA sequencing resolves molecular relationships among individual plant cells. *Plant Physiol.* **179**, 1444–1456 (2019).
 6. Denyer, T. *et al.* Spatiotemporal Developmental Trajectories in the Arabidopsis Root Revealed Using High-Throughput Single-Cell RNA Sequencing. *Dev. Cell* **48**, 840-852.e5 (2019).
 8. Alexandre, C. M. *et al.* Complex relationships between chromatin accessibility, sequence divergence, and gene expression in arabidopsis thaliana. *Mol. Biol. Evol.* **35**, 837–854 (2018).
 19. Bhosale, R. *et al.* A spatiotemporal dna endploidy map of the arabidopsis root reveals roles for the endocycle in root development and stress adaptation. *Plant Cell* **30**, 2330–2351 (2018).
 23. Sullivan, A. M. *et al.* Mapping and dynamics of regulatory DNA and transcription factor networks in A. thaliana. *Cell Rep.* **8**, 2015–2030 (2014).
 42. Pliner, H. A. *et al.* Cicero Predicts cis-Regulatory DNA Interactions from Single-Cell Chromatin Accessibility Data. *Mol. Cell* **71**, 858-871.e8 (2018).

REVIEWER COMMENTS

Reviewer #1 (Remarks to the Author):

The authors show predicted gene expression for two cherry-picked examples of late endodermis genes from the Shulze et al. data and largely don't address the other lines of evidence regarding the concern I raised about the endodermis developmental trajectory.

What would fully assuage my concern at this point is performing a correlation analysis (for all genes) between the predicted scRNA-seq expression for the Dorrity et al. endodermis clusters (endo_1, endo_2, and endo_3) versus existing developmental gene expression data from one or more of the following sources: 1) the Shulze endodermis clusters, 2) the Shahan et al. (<https://doi.org/10.1101/2020.06.29.178863>) endodermis stages, or 3) data from FACS-sorted endodermis isolated from the different root developmental zones (if this exists in the literature). Add this analysis to a supplementary figure (e.g., as a heatmap in the style of Supplementary Figure 3A but for correlation rather than proportion). Assuming the developmental trajectories agree, this would fully assuage my concerns about the endodermis developmental trajectory orientation. If they do not agree, then address this disagreement directly in the Discussion. The underlying issue is that the developmental progression scores used by the paper seem solely to be based on the general premise that later stage cells express (or have chromatin open at) fewer unique genes, but this does not appear to be benchmarked directly against any data from endodermis cells of specific developmental stages to validate its use as a metric.

Reviewer #2 (Remarks to the Author):

The authors have clarified how the endoreduplication scores were calculated. As I understand it now, 2N, 4N and 8N scores for each cell and cell cluster are calculated by taking the top-250 'marker' genes of Bhosale et al. (2018) peaking in 2N, 4N and 8N, respectively, calculating the average of normalized log expression values for each of these marker gene sets in the cell or cluster concerned (2N, 4N and 8N signature), and then calculating the ratio of e.g. the 8N and 2N signature to estimate whether or not the cell (cluster) is endoreduplicated. The 8N/2N measure used by the authors could technically call (real) 4N cells as either 2N or 8N, as it is unclear whether real 4N cells have a higher 2N or 8N signature based on the marker gene sets used by the authors (the only thing that can reasonably be assumed is that 4N cells have a higher 4N signature). In this sense, if it turns out 4N cells are inferred as 2N, the 8N/2N may indeed be a conservative measure of endoreplication (and otherwise it would be less conservative but still adequate). I still think the endoreplication measures used by the authors are tricky, but as they seem to produce reasonable results, I'm ok with them.

Reviewer #5 (Remarks to the Author):

The authors have addressed my concerns.

Reviewer #6 (Remarks to the Author):

I agree with reviewer 1 that there appears to be some inconsistency between the scATACseq data-based classification of the three different states of endodermal differentiation in comparison to what is known about this process. The mentioning that wax/suberin related genes are early markers is not correct, this was also shown by Naseer et al. (2012). Most of these genes come-up later and in a patchy pattern and gradually are expressed in most endodermal cells. Although the authors make a good point that accessibility does not necessarily shows a good correlation to gene expression, some

of the evidence they provide in order to support their classification is not helping in proving their point. As described by Kamiya et al., (2015), it is clear that MYB36 is expressed already in the early endodermis and that its expression appears to increase as the endodermis differentiates. This would fit with their prediction that subtype 3 has more MYB36 transcripts.

All 5 CASP genes are induced at the same time, when the endodermis starts to build its CS, which is still a relative early state of endodermal differentiation. At this point, SGN1 and ESB1 are also expressed, so these cannot be classified as late genes. However, their expression does increase during the differentiation process. I can confirm this as I have personally worked with all these genes. In addition, also other genes required for a normal CS, UCC1 and UCC2 are expressed in this zone (Reyt et al., 2020).

As a side note, since the authors mentioned MSL4, MSL6 and MSL10, they might find it interesting to know that these genes are mostly expressed in the root apical meristem zone (endodermis and cortex) and gradually go down in expression as the endodermis differentiates. They are thus expressed much earlier than the CASPs. This is unpublished data from my lab, but I just wanted to mention this to highlight that it indeed difficult to use accessibility data as a clear proxy for gene expression. Again, this is also mentioned by the authors in their response to reviewer 1. The authors have also added new text explaining this discrepancy and that this is a long-standing debate in the field. The reported data by Kamiya et al. (2015) showing that MYB36 is already expressed in undifferentiated endodermal cells and that subsequently its expression appears to increase, again correlates with the prediction of highest expression of MYB36 in subtype 3.

Thus, in general I think that the classification of the 3 endodermal subtypes proposed by the authors holds true, but in my opinion they might not have used the best candidate genes (ESB1 and SGN1) to make this point, suberin-related genes such as GPAT5 might be better. I think it would be good for the authors to really drive home the point that the correlation between accessibility and expression data is very weak in most cases and should be carefully interpreted. In that light it is good to overlay the scATACseq data with scRNAseq data.

Joop Vermeer

Dear Dr. Chuanfu An,

Please find our revisions below. We have resolved the remaining requests of our reviewers, which focused primarily on the endodermis sub-clusters and their developmental relationship. We present now five lines of evidence to confirm the reported developmental progression of the endodermis sub-clusters. Since we have addressed the remaining concerns to the best of our ability, , in a third round of revisions after our initial submission on February 3rd 2020, we kindly ask you to make a decision at this time. Thank you for your help in improving the clarity and the content of this manuscript.

Sincerely, Josh Cuperus

REVIEWER COMMENTS

Reviewer #1 (Remarks to the Author):

The authors show predicted gene expression for two cherry-picked examples of late endodermis genes from the Shulse et al. data and largely don't address the other lines of evidence regarding the concern I raised about the endodermis developmental trajectory.

What would fully assuage my concern at this point is performing a correlation analysis (for all genes) between the predicted scRNA-seq expression for the Dorrity et al. endodermis clusters (endo_1, endo_2, and endo_3) versus existing developmental gene expression data from one or more of the following sources: 1) the Shulse endodermis clusters, 2) the Shahan et al. (<https://doi.org/10.1101/2020.06.29.178863>) endodermis stages, or 3) data from FACS-sorted endodermis isolated from the different root developmental zones (if this exists in the literature). Add this analysis to a supplementary figure (e.g., as a heatmap in the style of Supplementary Figure 3A but for correlation rather than proportion). Assuming the developmental trajectories agree, this would fully assuage my concerns about the endodermis developmental trajectory orientation. If they do not agree, then address this disagreement directly in the Discussion.

We thank the reviewer for clearly outlining the requested analyses. In completing them, we found that our proposed developmental trajectory agrees with two of the reference datasets indicated by the reviewer. Boxplots showing the signature scores computed from Shulse *et al.* (as in suggested analysis 1) and a heatmap of correlation with FACS-sorted endodermis cells from root developmental zones (as in suggested analysis 3) are now included as Supplementary Figures 5F and 5G (reproduced below).

The results of the suggested analysis of the Shulse et al endodermis genes is most clear: the signature for early expressed genes is highest in the "endo 1" sub-type.

As a reminder, the signature scores in **Figure S5F** were computed using normalized expression values for each gene, producing a log-scale average expression value (signature) for each set (early, middle, late) of genes provided in Shulse et al. We did not use the data in Shahan et al to compute signatures as this seemed redundant with the approach suggested with the Shulse et al data.

We have added text to the respective figure legend to explain the analyses suggested by the reviewer:

“(F) Signature scores computed from early (n = 306 genes), mid (n = 358 genes), and late (n = 134 genes) from a previous scRNA-seq study analyzing endodermis development.³ Predicted expression of the early signature was highest in endodermis sub-type 1; the middle signature was highest in endodermis sub-type 2; the late signature was also highest in endodermis sub-type 2, but was lowest in endodermis sub-type 1. (G) Heatmap showing Pearson correlation coefficients from each endodermis sub-type’s (rows) predicted expression to each stage of endodermis cells defined by FACS-sorted cells (columns along x-axis, from early to late).¹ While all endodermis sub-types appeared to show greatest correlation to endodermis slice 8, the highest correlations for the earliest FACS-sorted endodermis stages was with endodermis sub-type 1.”

At this point, we provide several lines of evidence, beyond individual marker genes, to support the orientation of the endodermis developmental trajectory in our three scATAC clusters:

- (1) The early “endo 1” sub-type shows the highest highest expression of the “early” endodermis signature according to Shulse *et al*.
- (2) The “endo 1” sub-type shows the lowest expression of the “late” endodermis signature from Shulse *et al*.

- (3) The “endo 1” sub-type shows a greater correlation with the earliest FACS-sorted endodermis cells from Brady *et al.* compared to “endo 2” or “endo 3” sub-types.
- (4) The “endo 3” sub-type shows the highest level of the developmental progression metric, while the “endo 1” sub-type shows the lowest level (metric benchmarked with Brady *et al.* data, please see below), using both scRNA and scATAC data.
- (5) The “endo 3” sub-type has a higher predicted level of endoreduplication than the “endo 1” sub-type.

The underlying issue is that the developmental progression scores used by the paper seem solely to be based on the general premise that later stage cells express (or have chromatin open at) fewer unique genes, but this does not appear to be benchmarked directly against any data from endodermis cells of specific developmental stages to validate its use as a metric.

We were disheartened to read this comment, as we thought we had taken care to explain this analysis in the initial submission of the manuscript (**Figure S5B, reproduced below**). However, we recognize that we failed to sufficiently explain the analysis and figure in the text; in particular, we failed to highlight the importance of benchmarking the developmental progression score against an orthogonal assay of developmental progression. We have now added a much more explicit reference to this figure in the main text:

“We benchmarked the developmental progression score directly against data from endodermis cells of specific developmental stages and found a strong relationship between loss in transcriptional diversity and endodermis developmental stage (**Figure S5B**).²”

In short, we visualized the developmental progression score from individual endodermis cells that were assigned labels of endodermis developmental progression via correlation to FACS-sorted endodermis cells.^{1,2} Those cells with highest correlations to “late” endodermis cells (slices 9 – 12 from Brady *et al.*) had much greater developmental progression scores than those with highest correlations to “early” endodermis cells (slices 1 – 4).

Reviewer #2 (Remarks to the Author):

The authors have clarified how the endoreduplication scores were calculated. As I understand it now, 2N, 4N and 8N scores for each cell and cell cluster are calculated by taking the top-250 ‘marker’ genes of Bhosale et al. (2018) peaking in 2N, 4N and 8N, respectively, calculating the average of normalized log expression values for each of these marker gene sets in the cell or cluster concerned (2N, 4N and 8N signature), and then calculating the ratio of e.g. the 8N and 2N signature to estimate whether or not the cell (cluster) is endoreduplicated. The 8N/2N measure used by the authors could technically call (real) 4N cells as either 2N or 8N, as it is unclear whether real 4N cells have a higher 2N or 8N signature based on the marker gene sets used by the authors (the only thing that can reasonably be assumed is that 4N cells have a higher 4N signature). In this sense, if it turns out 4N cells are inferred as 2N, the 8N/2N may indeed be a conservative measure of endoreplication (and otherwise it would be less conservative but still adequate). I still think the endoreplication measures used by the authors are tricky, but as they seem to produce reasonable results, I’m ok with them.

This explanation accurately captures how the endoreduplication signatures were computed, and our interpretations of 8N vs 4N levels in these cells.

Reviewer #5 (Remarks to the Author):

The authors have addressed my concerns.

Reviewer #6 (Remarks to the Author):

I agree with reviewer 1 that there appears to be some inconsistency between the scATACseq data-based classification of the three different states of endodermal differentiation in comparison to what is known about this process. The mentioning that wax/suberin related genes are early markers is not correct, this was also shown by

Naseer et al. (2012). Most of these genes come-up later and in a patchy pattern and gradually are expressed in most endodermal cells. Although the authors make a good point that accessibility does not necessarily shows a good correlation to gene expression, some of the evidence they provide in order to support their classification is not helping in proving their point. As described by Kamiya et al., (2015), it is clear that MYB36 is expressed already in the early endodermis and that its expression appears to increase as the endodermis differentiates. This would fit with their prediction that subtype 3 has more MYB36 transcripts.

All 5 CASP genes are induced at the same time, when the endodermis starts to build its CS, which is still a relative early state of endodermal differentiation. At this point, SGN1 and ESB1 are also expressed, so these cannot be classified as late genes. However, their expression does increase during the differentiation process. I can confirm this as I have personally worked with all these genes. In addition, also other genes required for a normal CS, UCC1 and UCC2 are expressed in this zone (Reyt et al., 2020).

As a side note, since the authors mentioned MSL4, MSL6 and MSL10, they might find it interesting to know that these genes are mostly expressed in the root apical meristem zone (endodermis and cortex) and gradually go down in expression as the endodermis differentiates. They are thus expressed much earlier than the CASPs. This is unpublished data from my lab, but I just wanted to mention this to highlight that it indeed difficult to use accessibility data as a clear proxy for gene expression. Again, this is also mentioned by the authors in their response to reviewer 1. The authors have also added new text explaining this discrepancy and that this is a long-standing debate in the field. The reported data by Kamiya et al. (2015) showing that MYB36 is already expressed in undifferentiated endodermal cells and that subsequently its expression appears to increase, again correlates with the prediction of highest expression of MYB36 in subtype 3.

Thus, in general I think that the classification of the 3 endodermal subtypes proposed by the authors holds true, but in my opinion they might not have used the best candidate genes (ESB1 and SGN1) to make this point, suberin-related genes such as GPAT5 might be better.

The *ESB1* and *SGN1* genes were useful examples to make a point in the revision process: one or two genes are not sufficient to order the developmental trajectory of cell clusters generated from scATAC-seq data. Nevertheless, expression patterns of individual marker genes are typically inferred through multiple, often orthogonal methods, making marker genes very useful for cell type annotation. We agree that *GPAT5* would be a useful gene to examine; however, this gene was not represented in our dataset, consistent with the sparseness of single-cell expression data.. We examined another root single-cell dataset for *GPAT5* expression and found it was also not captured (<https://www.zmbp-resources.uni-tuebingen.de/timmermans/plant-single-cell-browser/>).

I think it would be good for the authors to really drive home the point that the correlation between accessibility and expression data is very weak in most cases and should be carefully interpreted. In that light it is good to overlay the scATACseq data with scRNAseq data.

We appreciate that this message of the paper is recognized; the tendency to conflate these two different measurements undersells their value as complimentary descriptions of gene regulation.

Joop Vermeer

References

1. Brady, S. M. *et al.* A High-Resolution Root Spatiotemporal Map Reveals Dominant Expression Patterns. *Science (80-.)*. **318**, 801–806 (2007).
2. Jean-Baptiste, K. *et al.* Dynamics of gene expression in single root cells of *A. thaliana*. *Plant Cell* **31**, tpc.00785.2018 (2019).

REVIEWERS' COMMENTS

Reviewer #1 (Remarks to the Author):

I thank the authors for directly addressing my remaining concern. No additional concerns.

Reviewer #6 (Remarks to the Author):

I have no further comments. The authors have adequately addressed my minor concerns and I believe they have also addressed the last comments of the other reviewers in a clear way.

Joop Vermeer